# Instruction Embedding: Latent Representations of Instructions Towards Task Identification

**Yiwei Li**[1][†], **Jiayi Shi**[1][†], **Shaoxiong Feng**[2], **Peiwen Yuan**[1], **Xinglin Wang**[1],
**Boyuan Pan**[2], **Heda Wang**[2], **Yao Hu**[2], **Kan Li**[1][‡]

[1] School of Computer Science, Beijing Institute of Technology
[2] Xiaohongshu Inc

## Abstract

Instruction data is crucial for improving the capability of Large Language Models (LLMs) to align with human-level performance. Recent research LIMA demonstrates that alignment is essentially a process where the model adapts instructions' interaction style or format to solve various tasks, leveraging pre-trained knowledge and skills. Therefore, for instructional data, the most important aspect is the task it represents, rather than the specific semantics and knowledge information. The latent representations of instructions play roles for some instruction-related tasks like data selection and demonstrations retrieval. However, they are always derived from text embeddings, encompass overall semantic information that influences the representation of task categories. In this work, we introduce a new concept, instruction embedding, and construct **I**nstruction **E**mbedding **B**enchmark (**IEB**) for its training and evaluation. Then, we propose a baseline **P**rompt-based **I**nstruction **E**mbedding (**PIE**) method to make the representations more attention on tasks. The evaluation of PIE, alongside other embedding methods on IEB with two designed tasks, demonstrates its superior performance in accurately identifying task categories. Moreover, the application of instruction embeddings in four downstream tasks showcases its effectiveness and suitability for instruction-related tasks[1].

## 1 Introduction

Large Language Models (LLMs) have demonstrated remarkable proficiency in generating responses capable of addressing specific tasks according to provided instructions. Initially pre-trained for wide-ranging capabilities, they are subsequently fine-tuned using instruction-following datasets to enhance their ability to align with human preferences. LIMA has proved that alignment can be viewed as a straightforward process in which the model just learns the style or format for interacting with users to solve particular problems, where the knowledge and capabilities have already been acquired during pre-training (Zhou et al., 2023).

Text embeddings play a crucial role in a variety of NLP tasks such as semantic textual similarity (Agirre et al., 2012; Cer et al., 2017; Marelli et al., 2014) and information retrieval (Mitra et al., 2017; Karpukhin et al., 2020). Similarly, as a type of text, the latent represent of instructions is also essential for many tasks like data selection for instruction tuning (Wu et al., 2023a) and prompt retrieval for in-context learning (Su et al., 2023). Previous studies (Gao et al., 2021; Wang et al., 2024) obtain text embeddings by directly taking the token vector from language models. However, when it comes

---

[†]Equal contributions.

[‡]Corresponding author.

[1]Our code and data have been released on `https://github.com/Yiwei98/instruction-embedding-benchmark`.

```
Sample1 - different tasks

• Tell me the main idea of this article.
• Tell me the gender of the author of this blog post.

Similarity with text embedding: 0.9943
Similarity with instruction embedding: -0.0254

Sample2 – similar tasks

• Create a poem with at least 5 lines, rhyming
  pattern aabb.
• Write a limerick based on the following noun.

Similarity with text embedding: 0.3239
Similarity with instruction embedding: 0.8287
```

(a)                    (b)

(c)

Figure 1: (a) Case about cosine similarity between instructions. Visualization of (b) text embeddings and (c) instruction embeddings. The same color indicates the same task category.

to the embeddings of instructions, the key focus should lie in identifying task categories rather than capturing overall semantic information. This is because, as mentioned earlier, instruction fine-tuning helps models learn how to interact with users across different tasks, rather than specific capabilities and knowledge imparted by the instructions. Therefore, task similarities is far more important than semantic similarities for instructions. Figure 1 (a) shows the case where traditional text embedding methods exhibit high overall semantic and syntactic similarity between two samples which actually represent completely different tasks, but low similarity when they represent similar task.

In this work, we propose a new concept called instruction embedding, a specialized subset of text embedding that prioritizes task identification for instructions over the extraction of sentence-level semantic information. We construct a new benchmark for instruction embedding training and evaluation. Different from previous text embedding benchmark that only considered the semantic textual similarity, IEB is labeled by task categories of instructions. Inspired by that key instruction words especially verbs are highlighted through instruction tuning (Wu et al., 2023b), we first extract verb-noun pairs to clarify category, then manually select and label instructions with other syntactic structures. Besides, we also conduct category merging and employ GPT-4 to generate complex samples to make the benchmark more robust. IEB totally contains 47k samples dispersed across more than 1k categories, which can also be used for embedding training and downstream tasks.

To stimulate language models to generate better instruction embedding, we propose a prompt-based baseline method PIE. It leverages the template to obtain instruction embeddings by directing the model's attention towards the task type represented by the instructions. Despite PIE demonstrating good practicality as it already performs well without training, we can further enhance it by fine-tuning the model on IEB with contrastive learning. As a widely used method for training embedding models, contrastive learning requires positive and hard negative samples to provide training signals, which are hard to extract. In our study, the explicit category information available in IEB enables the straightforward extraction of positive samples by directly selecting two instructions from the same category. We can further construct hard negative samples by selecting samples from categories that share identical verbs or nouns, enhancing the challenge of differentiation. Figure 1 shows that PIE can effectively distinguish whether two instructions refer to the same task cluster.

We evaluate PIE and other embedding baselines on IEB with instruction clustering and intention similarity tasks, which shows that PIE can largely outperform other baselines and precisely identify the task categories. We also conduct four downstream tasks, where the superior results demonstrate that the proposed instruction embeddings are more suitable for instruction-related tasks than traditional text embeddings.

## 2 The IEB Benchmark

We present instruction embedding benchmark, IEB, for assessing the quality of the latent representation of instructions. In contrast to current text embedding benchmarks that assess semantic similarity, the primary focus for the space of instruction embeddings is task differentiation based on the given instructions. Therefore, we annotate instructions with their respective tasks in IEB. We define task as follows: a task of an instruction is a category of activities or work that we expect the LLM to perform, which can be represented by a key phrase (mostly verb-noun phrases). The definition of task is not influenced by specific content or knowledge. For example, "writing an article" is a task, but the specific topic of the article is not important.

### 2.1 Data Extraction

For convenience and authenticity, we derive samples from established datasets. Specifically, we adopt three extensively recognized instruction-tuning datasets: DatabricksDolly (Conover et al., 2023), Alpaca data (Taori et al., 2023), and Self-instruct data (Wang et al., 2023). Labeling instructions entirely through manual effort or large language models will incur significant costs. Therefore, it is best to first conduct coarse-grained grouping and filtering based on rule-based policies. Wu et al. (2023b) proves that instruction fine-tuning enables models to recognize key instruction words, which leads to the generation of high-quality responses. Furthermore, it also encourages models to learn word-word relations with instruction verbs. Inspired by these two findings, we argue that verbs and other key words are crucial in identifying the task denoted by an instruction, where the types of them can be effectively determined through syntactic analysis. Thus, we employ the Berkeley Neural Parser[1] (Kitaev and Klein, 2018; Kitaev et al., 2019) for parsing the instructions.

After manual observation and considering the task category requirements, instructions can generally be divided into the following four groups through corresponding parsing tag recognizer:

**VP (VB+NN)** denotes verb phrase structure where the verb is closest to the root of the parse tree and directly links to noun. Instructions with this structure account for more than 80% of the total number before filtering. We categorize each instruction based on its verb-noun combination, identifying it as a specific task type, such as *write story* or *generate sentence*. After restoring the verb tense and singular form of nouns, we classify instructions with the same verb-noun combination into the same category. We plot the top most common root verbs and their direct noun objects in Figure 2.

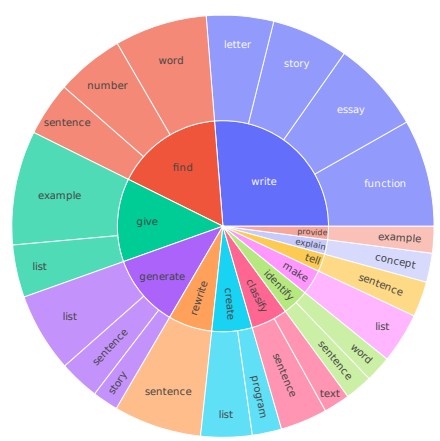

Figure 2: The verb-noun distributions in IEB.

**SBARQ** is direct question introduced by a wh-word or a wh-phrase. It can be divided into two main categories: knowledge-based questions led by six interrogative pronouns (e.g., what, when, where, ...) and math problems introduced by *what*. Unlike instructions in the VP (verb phrase) form, we define categories in the form of interrogative pronoun combing knowledge/math. This is because, considering they all involve asking about knowledge or math problems, further subdividing into noun categories is not very meaningful. For each category, we manually select around 50 samples.

**SQ** is inverted yes/no question. It can also be divided into two main categories: knowledge-based questions and task-oriented questions. Similarly, the task label is annotated as yes-no combing knowledge/task and we select around 50 samples for each category.

**Others** There are some other structures: verb phrase that lacks a direct connection to a noun and some rare cases which do not contain verbs, consisting only of noun phrases. We define these four

---

[1] https://parser.kitaev.io/

categories:(1) Verb-led knowledge questions. For example, knowledge clauses guided by *summarize* and *describe*. (2) Single verb for tasks, e.g., *translate*.(3) Verb-led mathematical problems. For example, math problem clauses guided by *multiply* and *simplify*.(4) Noun phrase for knowledge questions. For each type, we randomly select around 10-50 samples.

Finally, the annotated task categories cover the vast majority of the instruction data and are shown with examples in Table 1.

Table 1: Task categories with examples of IEB.

| Parsing Tag | Task Annotation | Examples |
|---|---|---|
| VP | verb + noun | Write an sessay about my favourite season. 
 Compose a song about the importance of computer science. |
| SBARQ | wh- + knowledge | What is the difference between machine learning and deep learning? 
 Why are matrices important in linear algebra? 
 How is a liquid chromatography differs from gas chromatography? 
 Who wrote the song House of Love? 
 When was the "No, They Can't" book released? 
 Where was 52nd International Film Festival of India held? |
| | what + math | What is the result when 8 is added to 3? 
 What is the value of (x - y)(x + y) if x = 10 and y = 15? |
| SQ | yes/no + knowledge | Was Furze Hill an established community in the 19th century? 
 Did Sir Winston Churchill win the Nobel Peace Prize? |
| | yes/no + task | Are the following two sentences grammatically correct? 
 Should this comma be included or omitted? |
| Others | verb + knowledge | Summarize the Challenger Sales Methodology for me. 
 Describe the Three Gorges Dam of China. |
| | verb | Translate "Bonjour" into English. 
 You need to translate ''I have been to Europe twice" into Spanish. |
| | verb + math | Multiply 12 and 11. 
 Simplify 2w+4w+6w+8w+10w+12. |
| | noun + knowledge | Short Summary about 2011 Cricket World Cup. 
 iPhone 14 pro vs Samsung s22 ultra. |

## 2.2 Data Synthesis

In instruction data, we discover some complex sentences, e.g., *Pretend you are a project manager of a construction company. Describe a time when you had to make a difficult decision.* Although they make up only a small portion of the dataset, they can serve as particularly challenging samples in the benchmark. However, due to their relative difficulty in identification, we employ GPT-4 to generate samples based on existing task category names, including verbs and corresponding nouns. The prompt and cases will be shown in the Appendix A.2. Subsequently, the generated compound instructions will be integrated into the categories.

## 2.3 Quality Control

**Automatic Filtering**  We find that low-frequency samples have a higher probability of being noisy, so we discard categories with fewer than 10 samples. Further, we employ GPT-4 to check whether samples belong to its annotated category. About 12.9% samples are filtered out during this process.The prompt is shown in Appendix A.3.

**Category Merging**  Considering that many verbs or nouns representing instructions are synonyms, e.g., *provide* and *give*, it would be inappropriate to classify them into different categories. We utilize WordNet [2] to extract the synonyms, where we merge every two categories where both nouns and verbs are synonyms or same words. Details is shown in Appendix A.3.

---

[2] https://wordnet.princeton.edu/

**Human Evaluation**   While we have highlighted the quantity and diversity of the data in IEB, the quality remains uncertain. To assess this, we randomly select 100 task categories and choose one instance from each. An expert annotator, who is a co-author of this work, then evaluate whether each instruction belongs to its annotated category. The instruction for judgement is the same as Automatic Filtering. The results indicate that 93% of the sample categories are accurate, showing that most of the annotated category labels are correct.

Table 2: Data statistics of IEB. EFT refers to embedding fine-tuning and IFT refers to instruction fine-tuning.

|     |       | Tasks | Samples |
|-----|-------|-------|---------|
| EFT | Train | 608   | 20814   |
|     | Test  | 145   | 3291    |
| IFT | Train | 600   | 21720   |
|     | Test  | 938   | 1336    |
|     | Total | 1353  | 47161   |

## 2.4  Statistics

After constructing and filtering, we collect totally 1353 task categories with 47161 samples. Given the large volume of data, the benchmark data can also be used for training and testing instruction embeddings and downstream tasks. Therefore, we have split it in a certain ratio, but it can be be adjusted freely as needed. The EFT (embedding fine-tuning) subset is designed to facilitate models in generating high-quality latent representations of instructions through embedding fine-tuning, which involves a supervised contrastive learning process based on our task labels (details on the embedding fine-tuning process can be found in Sections 3.2). The IFT (instruction fine-tuning) subset is constructed to evaluate the effectiveness of our instruction embeddings in downstream tasks, such as Data Selection for Instruction Tuning and Demonstration Retrieval (details available in Sections 4.3.1 and 4.3.2). Table 2 describes the statistics of the divided data. More statistics can be seen in Appendix A.4. Note that there is no overlap among the samples in the four parts, but the task categories in the training and test sets for IFT will overlap.

# 3  Instruction Embedding Method

Traditional text embeddings focus on capturing overall semantic information of text (Xu et al., 2023d). However, Zhou et al. (2023) and Wu et al. (2023b) demonstrate that the essence of instruction data lies in the tasks indicated by task words which are typically composed of a verb and a noun and specify the task action and the task domain (or object of action) respectively. Therefore, we propose instruction embedding method to capture task category information contained in instructions, rather than general semantic information.

## 3.1  Prompt-based Instruction Embedding

As mentioned above, guiding the model to generate embeddings that focus on task categories is critically important. LLMs have shown an impressive capacity to accomplish novel tasks solely by utilizing in-context demonstrations or instructions (Brown et al., 2020). Inspired by PromptBERT(Jiang et al., 2022), we present a prompt-based instruction embedding method (PIE) that employs a carefully designed prompt to guide the model in extracting the tasks embedded within given instructions. The hidden states of last input token will be represented for the embedding of instruction. The PIE-prompt is shown in Figure 15. Besides, a Semantic-prompt as shown in Figure 16 is also applied to model for comparison.

Table 3: Results of basic evaluation for instruction embedding. We conduct instruction clustering task and IIS test on each embedding method. Wiki refers to the train set of SimCSE (Xu et al., 2023c) and PromptBERT (Jiang et al., 2022), and semantic-prompt is shown in Figure 16.

| Method | | | ARI | CP | Homo | Silh | IIS-Spearman |
|---|---|---|---|---|---|---|---|
| **None-Fine-tuned** | | | | | | | |
| BERT | | | 0.3113 | 0.4853 | 0.6777 | 0.0792 | 0.5522 |
| BERT (semantic-prompt) | | | 0.2840 | 0.4524 | 0.6570 | 0.0936 | 0.5335 |
| BERT (PIE-prompt) | | | 0.2474 | 0.4038 | 0.6210 | 0.0706 | 0.4724 |
| Llama | | | 0.1813 | 0.3151 | 0.5439 | 0.0995 | 0.1565 |
| Llama2 (semantic-prompt) | | | 0.4238 | 0.5947 | 0.7549 | 0.1298 | 0.5893 |
| Llama2 (PIE-prompt) | | | 0.4814 | 0.6305 | 0.8014 | 0.1611 | 0.7189 |
| Vicuna | | | 0.1198 | 0.2859 | 0.4828 | 0.0934 | 0.1211 |
| Vicuna (semantic-prompt) | | | 0.1871 | 0.3145 | 0.5133 | 0.1081 | 0.6934 |
| Vicuna (PIE-prompt) | | | 0.5305 | 0.6633 | 0.8242 | 0.1732 | 0.7534 |
| **Unsupervised Fine-tuned** | | | | | | | |
| Wiki | w/o prompt | Llama2 | 0.3306 | 0.4877 | 0.6891 | 0.2185 | 0.1714 |
| | | BERT | 0.4741 | 0.6187 | 0.7741 | 0.1225 | 0.7460 |
| | semantic-prompt | Llama2 | 0.1776 | 0.3087 | 0.5412 | 0.0818 | 0.1476 |
| | | BERT | 0.3371 | 0.5084 | 0.6974 | 0.1161 | 0.6804 |
| **Supervised Fine-tuned with hard negative sampling** | | | | | | | |
| EFT-train | w/o prompt | Llama2 | 0.7541 | 0.8469 | 0.9143 | 0.3608 | 0.6038 |
| | | BERT | 0.8837 | 0.9392 | 0.9695 | 0.4574 | 0.8436 |
| | semantic-prompt | Llama2 | 0.8651 | 0.9204 | 0.9619 | 0.4542 | 0.8433 |
| | | BERT | 0.8876 | 0.9377 | 0.9683 | 0.4946 | **0.8450** |
| | PIE-prompt | Llama2 | **0.9125** | 0.9432 | 0.9697 | 0.4803 | **0.8450** |
| | | BERT | 0.8974 | **0.9453** | **0.9721** | **0.5180** | 0.8446 |

## 3.2 Embedding Fine-tuning

We further fine-tune PIE-model on EFT-train set following the contrastive learning (CL) framework in SimCSE (Gao et al., 2021), where we replace the dropout-based positive sample pairs construction method with a method based on instruction task labels from EFT-train.

Formally, let $\mathcal{D} = \{\mathbf{t}_i\}_{i=1}^{|\mathcal{D}|}$ denotes EFT-train, where each $\mathbf{t}_i = \{t_{i1}, ..., t_{|\mathbf{t}_i|}\}$ represents a specific task category in $\mathcal{D}$, and each $t_{ij}$ is an instruction instance from $\mathbf{t}_i$. During training, we take a cross-entropy objective with in-batch negatives (Chen et al., 2017; Henderson et al., 2017). For a given instruction $t_{ij}$, we randomly sampled $t_{ik}$ from $\mathbf{t}_i$ where $j \neq k$ to make up a task-related instruction pair. Let $h_{ij}$ and $h_{ik}$ denote the embeddings of $t_{ij}$ and $t_{ik}$, the learning objective for $(t_{ij}, t_{ik})$ with a mini-batch of N pairs can be formulated as Eq 1

$$\ell_i = -log \frac{e^{sim(h_{ij}, h_{ik})}/\tau}{\sum_{m=1}^{N} e^{sim(h_{ij}, h_{mk'})/\tau}} \tag{1}$$

where $\tau$ is the temperature hyperparameter and $sim(h_1, h_2)$ is the cosine similarity $\frac{h_1^T h_2}{||h_1|| \cdot ||h_2||}$.

Hard negative sampling has been widely adopted in CL (Schroff et al., 2015). In this paper, we propose a hard negative sampling strategy based on verb-noun style instruction task labels: for an instruction $t_{ij}$ whose task category is a verb-noun pair $(v_i, n_i)$, another instruction $t_{i'j'}$ whose task category is either $(v_i, n_{i'})$ or $(v_{i'}, n_i)$ is considered as a hard negative sample of $t_{ij}$. When searching for hard negative samples, we prioritize samples with the same verb but different nouns.

# 4 Experiment

## 4.1 Experimental Setup

Based on IEB benchmark, we introduce instruction clustering task (ICT) and instruction intention similarity (IIS) test to evaluate instruction embeddings. ICT aims to accurately group instructions from different tasks. Specifically, instruction clustering is conducted using k-means clustering based on embeddings of given instructions, where $k$ is predefined and its value equals to the number of task categories in EFT-test (i.e. $k = 145$ here). We utilize metrics such as Adjusted Rand Index (ARI) (Hubert and Arabie, 1985), Clustering Purity (CP) (Schütze et al., 2008), Homogeneity Score (Homo) (Rosenberg and Hirschberg, 2007) and Silhouette Score (Silh) (Rousseeuw, 1987) for evaluation. IIS test is designed to align with STS (Agirre et al., 2012) task. The IIS test set is derived from IFT-train set. First, we randomly sample 1.5k instruction pairs of the same task from IFT-train set and label them as 1. Next, we sample another 1.5k pairs, labeling them as 1 if the task categories matched, otherwise 0. This resulted in a rough 1:1 ratio of samples labeled 1 to those labeled 0[3]. During testing, we calculate cosine similarity of the instruction embeddings for each pair, and compute the Spearman value with the labels across the entire dataset.

We implement our PIE method with Llama2 (Touvron et al., 2023b) and BERT (Devlin et al., 2019) separately. For all BERT-based embedding methods, we take the hidden state of [CLS] token from the last layer as instruction embedding. For all Llama2, we first conduct preliminary experiments to select best pooling method and prompt. According to the results, we utilize the average of last token hidden states across last 2 layers as the instruction embedding and choose the prompt. Details of this preliminary experiment can be found in Appendix C.

We evaluate the instruction task representation capability of baseline models and compare their performance with our PIE and corresponding supervised fine-tuning method. The baselines are as follows:

**None-Fine-Tuned baselines** We employ Llama2, Vicuna-7b-v1.5 (Zheng et al., 2023) and BERT to obtain instruction embeddings with three prompts: no prompt, semantic-prompt, and PIE-prompt.

**Unsupervised Fine-Tuned baselines** Unsupervised SimCSE (Gao et al., 2021) and unsupervised PromptBERT (Jiang et al., 2022) are included as unsupervised fine-tuned baselines. To eliminate the impact of model scale, we also re-implement them with Llama2.

**Supervised Fine-Tuned baselines** We supervised fine-tune Llama2 and BERT as mentioned in Section 3.2. Detailed fine-tuning configurations can be found in Appendix D.

## 4.2 Results and Analyses

**Main Findings** The experimental results are shown in Table 3. For none-fine-tuned baselines, our PIE-Prompt guides LLMs to extract task categories of instructions, enabling them to achieve significant improvements in both ICT and IIS test compared to the same model without using prompt. BERT failed to benefit from PIE-prompt, which may due to its limited instruction following capability. Interestingly, Vicuna achieves better results than Llama2 with PIE-prompt despite performing worse when prompt is not used. This is because Vicuna has been enhanced its instruction following capability through instruction tuning, enabling it to better extract task-specific information under the guidance of the PIE prompt. Although Llama2 and Vicuna achieve better performance in none-fine-tuning setting with PIE prompt, BERT successfully bridges this gap and achieves comparable or even better results after supervised fine-tuning on EFT-training. Additionally, for both Llama2 and BERT, although the performance gap between models using PIE-prompt and those using semantic-prompt or no prompt significantly narrows after supervised fine-tuning, models using PIE-prompt still outperform the others. This demonstrates that the guidance provided by PIE-prompt remains crucial even after supervised fine-tuning. To better illustrate the superiority of PIE and the impact of supervised fine-tuning, we visualize instruction embeddings of various models. The visualization analysis is presented in Appendix E.

---

[3]The IIS test set is not derived from EFT-test because each task in EFT-test mostly contains only 1 or 2 samples.

**Impact of Different Prompts**   To better understand the impact of different prompts, we print the outputs of each model under various prompts. We find that without using prompts, Llama2 tends to repeat the instruction, while Vicuna which has undergone instruction fine-tuning, will execute the instruction. This explains why Llama2 outperforms Vicuna with no prompts since Llama2 retains more original instruction information in its output. When prompts are added, the models behavior are guided, enabling them to extract instruction information according to the prompt. However, when using semantic prompts, models focus more on analyzing instruction semantic information rather than task categories. Consequently, model performance with semantic prompts is not as good as those with PIE prompts. The model inference examples can be found in Appendix F.

**Ablation Studies**   We conduct ablation studies on hard negative sampling strategy. We compare the performance of supervised fine-tuned models with and without hard negative sampling on embedding clustering task and IIS test, the results are shown in Figure 3. After removing hard negative sampling, the performance of models using different prompts all show a decline on embedding clustering task and IIS test. Our hard negatives are constructed through overlap of verb or noun, which helps eliminate the short-cut of distinguishing positives and negatives by

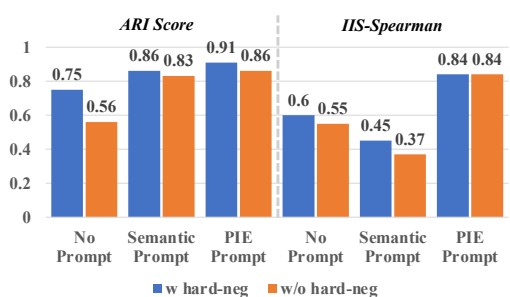

Figure 3: Results of ablation studies.

word overlap. This allows the model to better focus on the relationship between instruction tasks of positive and negative samples during training.

## 4.3   Evaluation on Downstream Tasks

We conduct four downstream tasks for further evaluation. Our core objective is to validate that instruction embeddings are more suitable for instruction-related downstream tasks compared to traditional text embeddings that focus on the overall semantic information of sentences. Therefore, we select the best-performing model we produced for each type of embedding, i.e., fine-tuned PIE-Llama2 and Wiki fine-tuned Llama2.

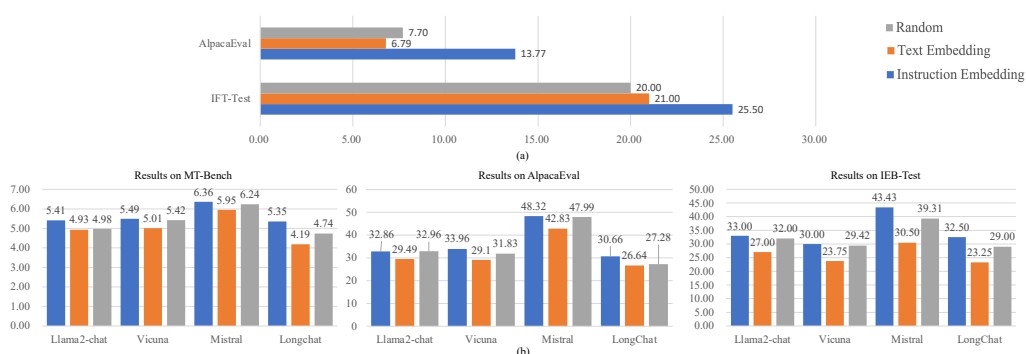

Figure 4: Results on (a) data selection for instruction tuning and (b) demonstrations retrieval for in-context learning.

### 4.3.1   Data Selection for Instruction Tuning

Following previous work (Wu et al., 2023a; Zhou et al., 2023), we design a data selection experiment based on embeddings for instruction diversity. First, we use k-means clustering to divide the IFT-train set into 600 clusters, and extract the closest samples to the clustering centers to achieve data compression. Then, we fine-tune Llama2 on that selected data. Training configurations can be found in Appendix D. We evaluate the performance on our IFT-test set and AlpacaEval (Li et al., 2023c). We use GPT-4 Turbo for judgment, and for IFT-test, its own output serves as the baseline for comparison. We take 5 runs for each setting and calculate the mean score. The result from Figure 4 (a) indicates

Table 4: Results of tiny benchmark. † denotes P-value $< 0.05$ and ‡ denotes $< 0.01$.

| Model | Instruction Embedding | | | Text Embedding | | | Random | | |
|---|---|---|---|---|---|---|---|---|---|
| | 10 | 50 | 100 | 10 | 50 | 100 | 10 | 50 | 100 |
| Llama2-chat | 18.40 | 6.89 | **3.17**† | 33.50 | **5.35** | 3.34 | **13.46** | 5.68 | 3.97 |
| Vicuna | **8.92**† | **3.76**‡ | **3.43**‡ | 13.22 | 8.56 | 3.61 | 11.53 | 5.88 | 4.61 |
| Mistral | 7.92‡ | **4.27**‡ | **2.14**‡ | **2.98** | 5.05 | 3.29 | 10.94 | 5.67 | 3.35 |
| Longchat | **7.76**‡ | **4.69**‡ | 3.70 | 28.82 | 4.74 | **3.47** | 12.07 | 6.11 | 4.22 |

that instruction embedding can be a better substitution of text embedding for enhancing the diversity of selected instructions. We additionally re-implement the data selection method DEITA with text embedding and instruction embedding separately, and the details can be found in Appendix K.

### 4.3.2 Demonstrations Retrieval

LLMs have shown remarkable in-context learning (ICL) capability (Patel et al., 2023; Yuan et al., 2024). Demonstrations related to the input instruction task are more conducive to model since task-related data are more similar in terms of format and content. Thus in this experiment, we select 2 most related instruction data by calculating cosine similarities from IFT-train set for each instruction in test set. The prompt template of ICL can be found in Appendix G. Similarly, we report evaluation results on IFT-test set and AlpacaEval with four models: Vicuna-7B-v1.5, Llama2-7B-chat (Touvron et al., 2023b), Mistral-7B-Instruct-v0.2 (Jiang et al., 2023), LongChat-7B-v1.5-32k (Li et al., 2023a). For random selection, we take 10 runs and report the mean score. The results are shown in Figure 4 (b), which demonstrates instruction embedding helps to select more task-related demonstrations and makes better ICL for LLMs.

### 4.3.3 Tiny Benchmark

Recently, some work has focused on testing models using fewer samples (Vivek et al., 2024; Polo et al., 2024). The primary goal is to select a more balanced tiny benchmark that can lead to more consistent performance compared to the original full benchmark. Similar to data selection for instruction tuning, this process can also be accomplished through clustering. We select 10, 50, and 100 test samples respectively, and compare the estimation error (%) in performance between the tiny and the original IFT-test benchmark. Following Vivek et al. (2024), we take 100 runs for each setting. The results in Table 4 indicates that instruction embedding can obtain a smaller estimation error by selecting more representative test samples.

### 4.3.4 Dataset Task Correlation Analysis

We analyze the correlation degree between instruction tasks across various open-source datasets through instruction embedding. Let $D_1, D_2$ denote two unique instruction datasets, for each instruction $t_i$ in $D_1$, we find its most relevant instruction $t'_{i'}$ in $D_2$ and take the average of $s_i$ (i.e. the similarity between $t_i$ and $t'_{i'}$) across $D_1$ (i.e. $\frac{\sum_{i=1}^{|D_1|} s_i}{|D_1|}$) as a measure of the extent to which the tasks in $D_1$ are encompassed in $D_2$. We conduct task correlation analysis across GSM8k (Cobbe et al., 2021), MATH (Hendrycks et al., 2021), MBPP (Austin et al.,

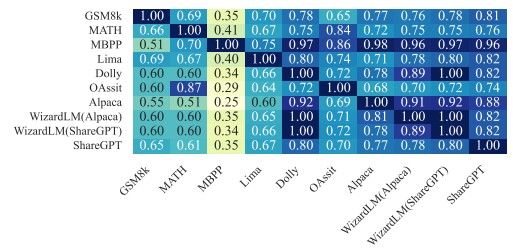

Figure 5: Correlation degree across various datasets through instruction embedding.

2021), Lima (Zhou et al., 2023), Dolly (Conover et al., 2023), OAssit (Köpf et al., 2023), Alpaca (Taori et al., 2023), WizardLM (WizardLM(Alpaca), WizardLM(ShareGPT)(Chiang et al., 2023a). As depicted in Figure 5, instruction embeddings succeed to distinguish between math tasks (GSM8k, MATH) and code tasks (MBPP). The correlation degree within math task datasets is significantly higher than the correlation degree between math task datasets and code dataset. Besides, larger and more general instruction datasets exhibit a more significant correlation with other datasets.

# 5 Related Work

**Text Embeddings**    Text embeddings are pivotal in NLP. They encapsulate overall semantic information and the quality of learned embeddings directly influences downstream tasks. Current research on text embeddings primarily focuses on sentence semantic modeling (Gao et al., 2021; Jiang et al., 2022; Li and Li, 2023). We argue that the essence of instructions lies in their task information and instruction embeddings should prioritize modeling task-specific information instead of emphasizing overall semantic information.

**Embedding Benchmark**    Semantic Textual Similarity (STS) tasks (Agirre et al., 2012; Cer et al., 2017; Marelli et al., 2014) are commonly employed to evaluate the quality of text embeddings, complemented with transfer tasks and short text clustering tasks (Conneau and Kiela, 2018; Xu et al., 2023d; Muennighoff et al., 2023) to further illustrate the superiority of learned sentence representations. However, previous benchmarks are not tailored to instruction corpora and primarily assess the semantic modeling abilities of text embeddings, rendering them less suitable for evaluating instruction embeddings.

**Instruction Tuning**    Instruction Fine-Tuning (IFT) is widely adopted to stimulate the instruction following capability of pre-trained LLMs. Early approaches for IFT focused on fine-tuning LLMs with large amounts of instruction data (Wang et al., 2022; Wei et al., 2022) manually aggregated from large NLP task collections (Longpre et al., 2023). With the development of generative language models, Wang et al. (2023) made their attempt to expand instruction data through synthetic data generation, inspiring the following works to evolve instruction data in this automated manner (Taori et al., 2023; Ding et al., 2023; Xu et al., 2023a). Zhou et al. (2023) proved that the quality and diversity of instruction data are significantly more critical than its sheer quantity, motivating recent efforts in instruction data selection to remove unnecessary IFT training costs by eliminating low-quality and redundant data. Quality-based data selection methods typically employ a quality evaluator to predict the quality scores of each instruction sample which are further used to select instruction data Chen et al. (2023); Li et al. (2023b). Diversity-based data selection methods aims to maximize the distance between selected instruction data which are measured by their embeddings Wu et al. (2023a); Liu et al. (2024). However, due to the lack of instruction embedding, previous works relied on semantic embedding which fails to emphasize the task-specific information of instructions data.

# 6 Conclusion

We introduce the concept of instruction embedding, which prioritizes task identification over traditional sentence-level semantic analysis. Alongside this, we release the publicly available IEB benchmark for evaluating and further training instruction embeddings. To ensure instruction embeddings focus more on task specifics, we propose a prompt-based approach for generating instruction embeddings, applicable in both learning-free and supervised fine-tuning settings. It has been demonstrated on two basic evaluation tasks and four downstream tasks that instruction embedding is superior for instruction-related tasks. The introduction of instruction embedding, along with the IEB benchmark and the PIE method, plays a crucial auxiliary role in instruction-related tasks for large language models.

# 7 Limitations

One limitation of our approach is that, by not relying entirely on manual labeling or verification, not all the data is guaranteed to be of high quality. Manual validation results indicate that 93% of the sample categories are accurate, leaving a small portion that may still contain noise. Additionally, we have not addressed multi-step instructions, where several serialized tasks are embedded within a single instruction, as no such cases were manually identified in the selected dataset, and therefore, these samples were neither handled nor supplemented. Lastly, the three popular instruction datasets we selected consist solely of single-turn interactions, meaning that the benchmark does not include multi-turn samples.

## Acknowledgments and Disclosure of Funding

This work is supported by Beijing Natural Science Foundation (No.4222037, L181010).

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

# A  Appendix about Benchmark

## A.1  Data Availability

**Dataset**: The whole benchmark along with four split parts can be found in `https://github.com/Yiwei98/instruction-embedding-benchmark`.

**Code**: The code for experiments can be found in `https://github.com/Yiwei98/instruction-embedding-benchmark`.

## A.2  Details about Data Synthesis

The prompt for employing GPT-4 to generate samples based on task category names is shown in Figure 6. We randomly selected 30% existing task categories and generate 3 samples for each category. After filtering, we obtained a total of 633 synthetic samples.

Generate an instruction represents the {task category} task, which contains two sentences. Note that the second generated sentences must contain the task word.

Figure 6: The prompt for generating the complex instructions.

Here are some generated examples:

Table 5: Examples of generated complex instructions.

| Task category | Examples |
|---|---|
| Classify Animal | You are a biologist studying a new species discovered in the Amazon rainforest. Classify the animal based on its characteristics, habitat, and behavior. |
| Generate Rap | Imagine you are a famous rapper who's known for his/her unique style. Generate a rap verse that showcases your creativity and lyrical prowess. |
| Give Title | You have written an article about the impact of social media on mental health. Give a title to your article that will reflect the content of your article. |
| Make Poem | Imagine you are sitting by a serene lake during a beautiful sunset. Make a poem that captures this tranquil moment and the emotions it evokes. |

## A.3  Details about Quality Control

The prompt for employing GPT-4 to check whether instructions belong to its annotated category is shown in Figure 7.

Check if the given instruction represents the {task category} task. Instruction: {instruction}. Please answer 'yes' or 'no'.

Figure 7: The prompt for generating the complex instructions.

For category merging, we will provide additional details about the merging procedure. Firstly, we select every two categories where both nouns and verbs are synonyms or same words. Then we calculate the cosine similarities of each pair of them by using word embeddings. For two categories where the values of both nouns and verbs pairs are above 0.5, we directly merge them as one category. For categories with values between 0.3 and 0.5, we use GPT-4 to determine whether they describe the same task. If they do, we merge them. For those below 0.3, we directly discard the merge. The prompt for this process is shown in Figure 8.

Are {task1} and {task2} represent the same task for instruction?. Please answer 'yes' or 'no'.

Figure 8: The prompt for generating the complex instructions.

## A.4 More Statistics

Besides the dataset partitioning, we provide more information about the statistics of proposed benchmark. We present the distribution of the number of instructions per category in Figure 9. Please note that for categories with more than 100 samples, we randomly retained only 100. Additionally, Figures 10 through 14 provide a more detailed view of the verb-noun distributions, where it is clear that there is no category overlap between EFT and IFT, but there is some overlap between the training and test sets within IFT.

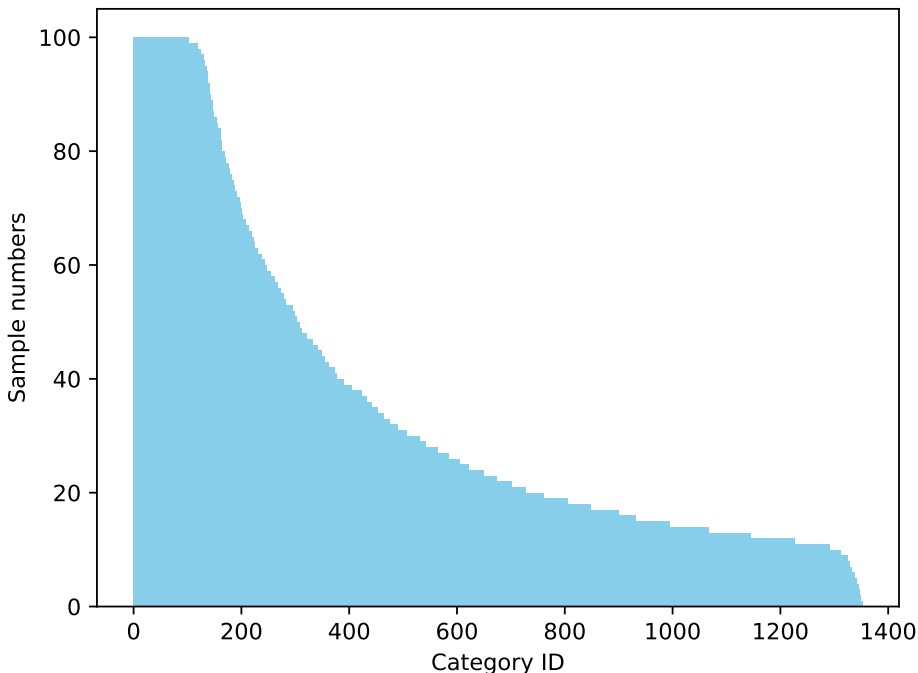

Figure 9: Distribution of the number of instructions per category.

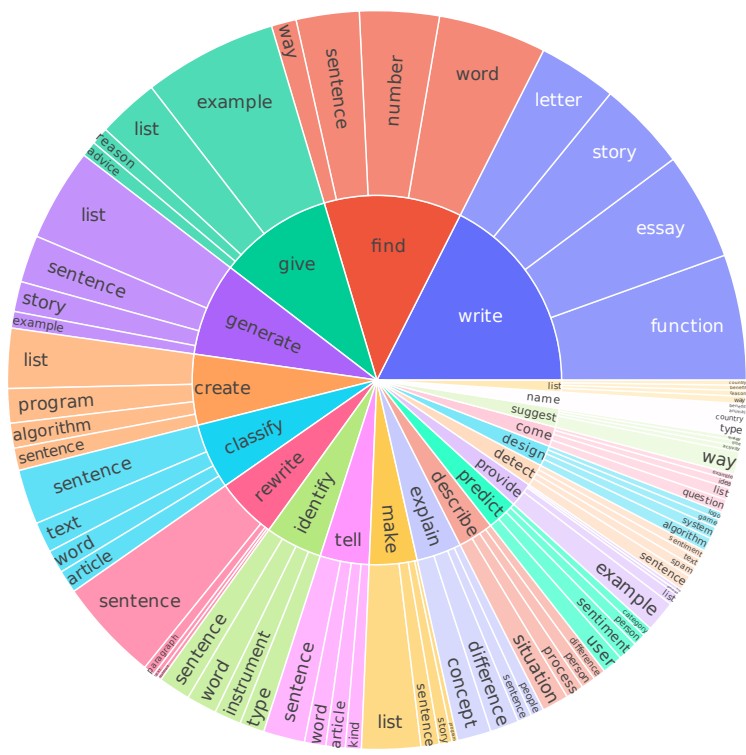

Figure 10: Verb-noun distributions of whole benchmark.

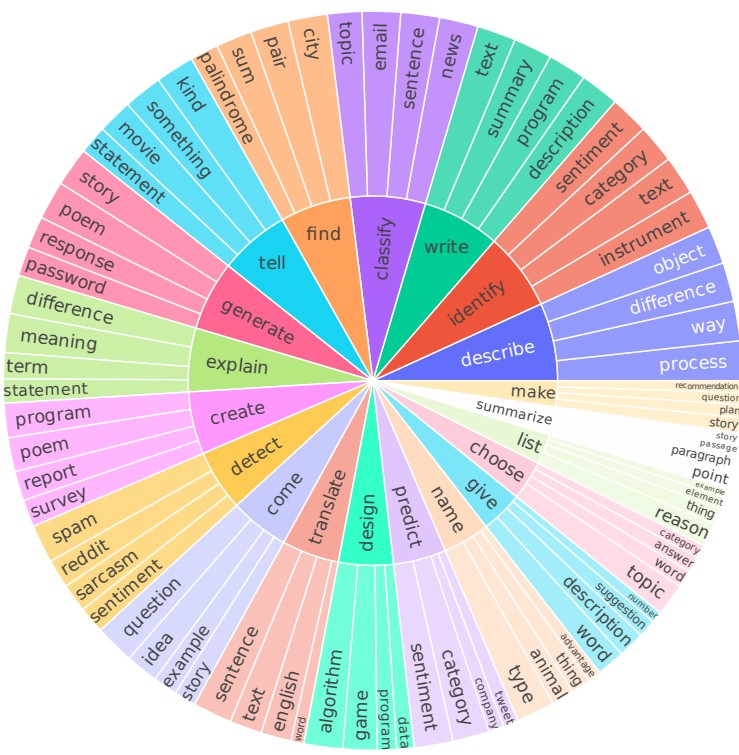

Figure 11: Verb-noun distributions of EFT-train.

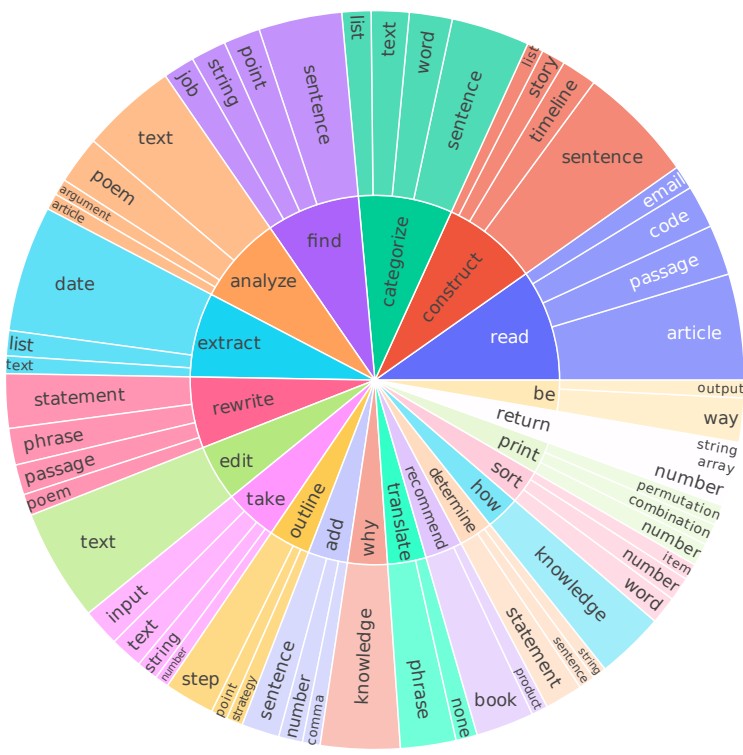

Figure 12: Verb-noun distributions of EFT-test.

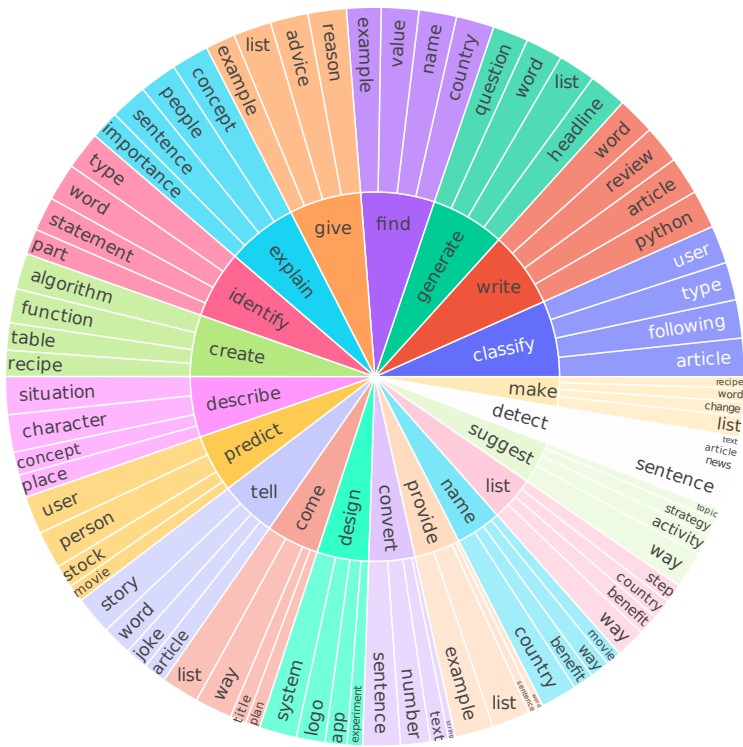

Figure 13: Verb-noun distributions of IFT-train.

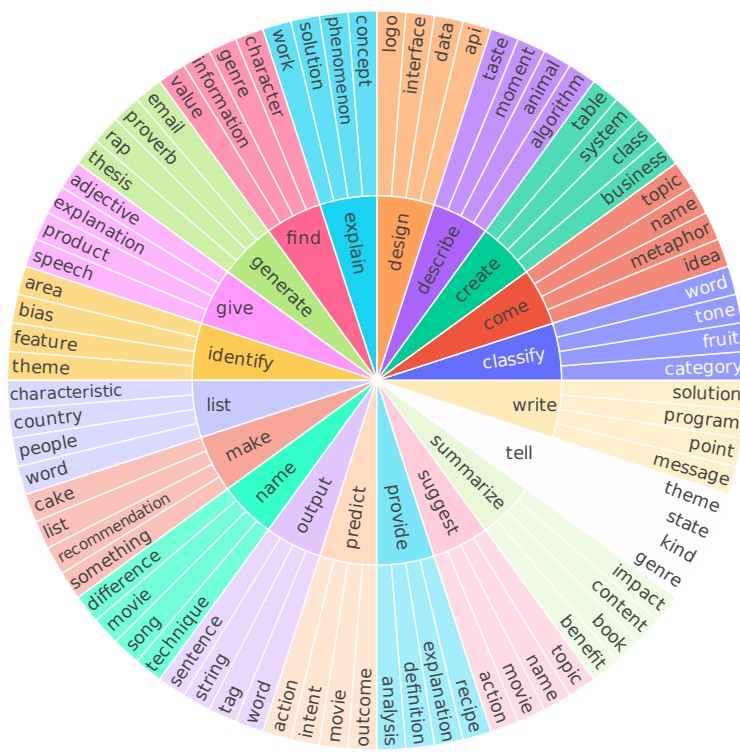

Figure 14: Verb-noun distributions of IFT-test.

## B  Prompts

**PIE Prompt**  The PIE Prompt is shown in Figure 15. Inspired by Zhang et al. (2024), we combine the pretended chain of thought method and knowledge enhancement method in this prompt, which effectively enhances the instruction task capturing capability of LLM. The prompt search preliminary experiment is shown in Appendix C.2.

The essence of an instruction is its task intention. With this in mind, given the instruction below:

{Instruction}

after thinking step by step, the task of the given instruction is:

Figure 15: The PIE Prompt.

**Semantic Prompt**  The semantic Prompt is shown in Figure 16.

This Sentence of {Instruction} means:

Figure 16: The Semantic Prompt.

## C  Preliminary Experiments

### C.1  Pooling Layer Selection

In LLM, the effectiveness and performance of extracting sentence representations across different hidden layers may vary. To systematically assess the semantic information and representation capabilities of various layers of Llama2[4], we employs pooling techniques on the last token hidden states at different layers and conduct corresponding evaluations. Specifically, we select the last hidden layer, last two hidden layers, middle hidden layer, and first and last hidden layers as pooling layers. The experimental results are shown in Table 6. We finally select the last two layers as pooling layers mainly due to its robustness. Although it does not achieve all the best results, it consistently maintains competitiveness against the best scores in each metric.

Table 6: Results of pooling layer selection experiment. For all pooling layers, we take the average pooling of last token hidden states in each chosen hidden layer as the instruction embedding.

| Layer | CP | ARI | Homo | Silh | IIS-Spearman |
|---|---|---|---|---|---|
| Last two | 0.1813 | 0.3151 | 0.5439 | 0.0995 | **0.1565** |
| Last one | **0.1868** | 0.3096 | **0.5466** | 0.1085 | 0.1414 |
| First-Last | 0.1825 | **0.3157** | 0.5450 | **0.1121** | 0.1413 |
| Mid | 0.1260 | 0.2446 | 0.4601 | 0.1321 | 0.1051 |

### C.2  Prompt Search

Prompt is a key part of our PIE . In this paper, we employed a manual approach to search for appropriate prompt: we first manually crafted several prompts, then, for each manually crafted prompt, we evaluated its effectiveness by the instruction clustering task. The human crafted prompts are shown in Table 7, and the results are presented in Table 8. According to the result, we select #5 template for further experiments.

---

[4]The model here is none-fine-tuned.

Table 7: Templates used in prompt search.

| Index | Template |
|---|---|
| #0 | Below is an instruction that describes a task
{instruction}
The task of the given instruction is: |
| #1 | The following instruction
{instruction}
wants you to: |
| #2 | Given the following instruction
{instruction}
please identify its task type: |
| #3 | What type of task does the following instruction represent?
{instruction} |
| #4 | Indentify the task category associated with the following instruction:
{instruction} |
| #5 | The essence of an instruction is its task intention. With this in mind, given the instruction below:
{instruction}
after thinking step by step, the task of the given instruction is: |

Table 8: Result of prompt search. Index refers to the template index in Table 7.

| Index | ARI | CP | Homo | Silh | IIS-Spearman |
|---|---|---|---|---|---|
| #0 | **0.4825** | **0.6308** | 0.7942 | **0.1672** | 0.6736 |
| #1 | 0.4233 | 0.5761 | 0.7504 | 0.1476 | 0.5897 |
| #2 | 0.3231 | 0.4959 | 0.6980 | 0.1340 | 0.5309 |
| #3 | 0.2512 | 0.4053 | 0.6227 | 0.1262 | 0.4054 |
| #4 | 0.2723 | 0.4108 | 0.6383 | 0.1175 | 0.3427 |
| #5 | 0.4814 | 0.6305 | **0.8014** | 0.1611 | **0.7189** |

# D  Additional Configuration

**Instruction Embedding Fine-tuning Experiment Configurations**  We complete each embedding fine-tuning on a single NVIDIA A100 GPU and adopt LoRA Hu et al. (2022) technique to fine-tune Llama2 7B[5] with lora-rank set to 32, lora-alpha set to 64, lora-dropout set to 0.05 and target modules set to ['q_proj','v_proj'][6]. During training, we set epochs to 1, batch size to 16, tokenize maxlength to 256. Following Gao et al. (2021), the temperature hyperparameter $\tau$ in Eq 1 is set to 0.05. Notably, to better focus on investigating the impact of our embedding train data on instruction embedding training, we remove the data augmentation methods in SimCSE during the embedding training process. Additionally, BERT refers to *bert=base-uncased*[7] and Vicuna refers to *vicuna-7b-v1.5*[8] unless otherwise specified.

**Configurations for Instructing Tuning.**  We complete instruction fine-tuning on 8 NVIDIA A100 GPU to fine-tune the LLM with the batch size set to 128 and the learning rate set to $2 * 10^{-5}$. The Alpaca-style template is applied to concatenate queries and responses during fine-tuning.

# E  Visualization Analysis

To better illustrate the superiority of PIE and the impact of supervised fine-tuning, we visualize instruction embeddings of various mdoels in Figure 17. It is evident that embedding fine-tuning

---

[5]https://huggingface.co/meta-llama/Llama-2-7b-hf

[6]https://huggingface.co/docs/peft/developer_guides/lora

[7]https://huggingface.co/google-bert/bert-base-uncased

[8]https://huggingface.co/lmsys/vicuna-7b-v1.5

successfully enhances the performance of both prompt-free models and PIE-models in terms of instruction clustering. This suggests that supervised instruction embedding fine-tuning aids in extracting task category more accurately from instructions. Additionally, fine-tuned PIE-models exhibits a more dispersed inter-class distribution and a more compact intra-class distribution than the fine-tuned prompt-free models, demonstrating the positive guiding effect of the prompt method on extracting task category information from instructions.

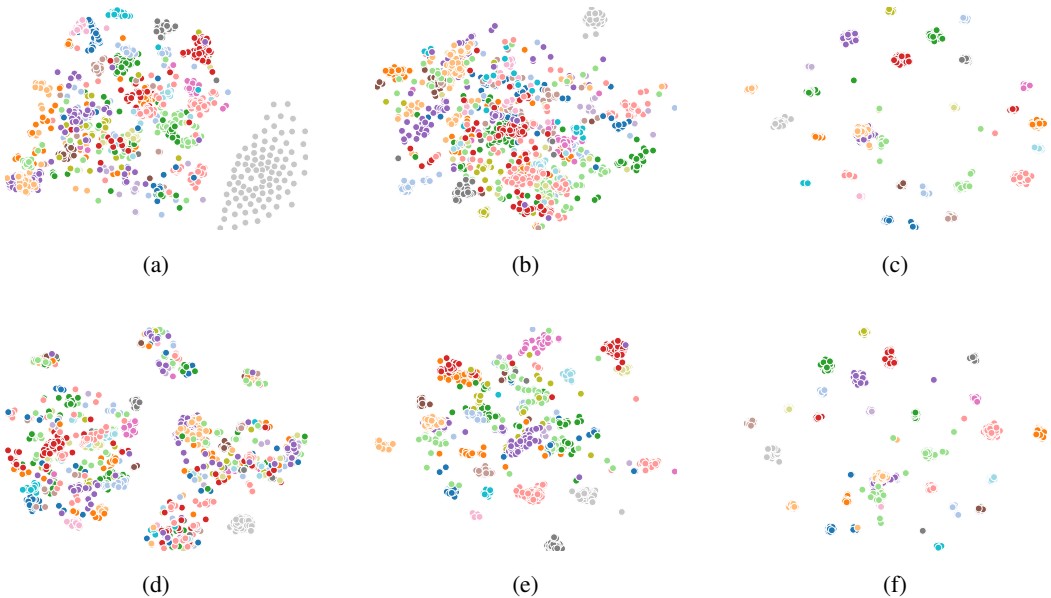

Figure 17: Embedding visualization: (a) BERT (b) BERT(pie-prompt) (c) Sup BERT(pie-prompt) (d) Llama2 (e) Llama2(pie-prompt) (f) Sup Llama2(pie-prompt)

# F  Model Inference Output Examples

In this section, we present the examples of model inference outputs under guidance of different prompts. The results of Llama[9] Touvron et al. (2023a), Llama2, Vicuna v1.3[10] Chiang et al. (2023a), Vicuna v1.5 are shown below. We remove the blank lines from the original model output and replaced the repeatedly generated content with ellipses (...).

## F.1  Model Inference Output Examples without Prompt.

**Instruction**
Count how many words are there in the paragraph. also count how many characters are there. output both numbers.
**LLaMA output**
<unk>Count how many words are there in the paragraph. also count how many characters are there. output both numbers.
Comment: @user3100112 I'm not sure what you mean by "output both numbers". Do you mean to print the numbers?
Comment: @user3100112 I've updated my answer to include the output.
…

(a)

**Instruction**
Count how many words are there in the paragraph. also count how many characters are there. output both numbers.
**LLaMA2 output**
Count how many words are there in the paragraph. also count how many characters are there. output both numbers.
I have a problem with the following code.
…

(b)

**Instruction**
Count how many words are there in the paragraph. also count how many characters are there. output both numbers.
**Vicuna-v1.3 output**
 Count how many words are there in the paragraph. also count how many characters are there. output both numbers.
Example:
Input: "The quick brown fox jumps over the lazy dog."
Output: 41 (words), 41 (characters)

(c)

**Instruction**
Count how many words are there in the paragraph. also count how many characters are there. output both numbers.
**Vicuna-v1.5 output**
 Count how many words are there in the paragraph. also count how many characters are there. output both numbers.

(d)

Figure 18: Model infer output examples (prompt-free).

---

[9]https://huggingface.co/yahma/llama-7b-hf
[10]https://huggingface.co/lmsys/vicuna-7b-v1.3

## F.2 Model Inference Output Examples with Semantic Prompt.

**Instruction**

Count how many words are there in the paragraph. also count how many characters are there. output both numbers.

**Semantic-Prompt LLaMA output**

<unk>This sentence of "Count how many words are there in the paragraph. also count how many characters are there. output both numbers." means: 1. count how many words are there in the paragraph. 2. count how many characters are there. output both numbers.

Comment: @user3580874 I'm not sure what you mean by "output both numbers". I'm not sure what you mean by "output both numbers". I'm not sure what you mean by "output both numbers". I'm not sure what you mean by "output both numbers". I'm not sure what you mean by "output both numbers". I'm not sure what you mean by "output both numbers". I'm not sure what you mean by "output both numbers". I'm not sure what you mean by "output both numbers". I'm not sure what you mean by "output both numbers". I'm not sure what you mean by "output both numbers".

…

(a)

**Instruction**

Count how many words are there in the paragraph. also count how many characters are there. output both numbers.

**Semantic-Prompt LLaMA2 output**

This sentence of "Count how many words are there in the paragraph. also count how many characters are there. output both numbers." means:

Count how many words are there in the paragraph. also count how many characters are there. output both numbers.

"Count how many words are there in the paragraph. also count how many characters are there. output both numbers." means:

Count how many words are there in the paragraph. also count how many characters are there. output both numbers. 1.

…

(b)

**Instruction**

Count how many words are there in the paragraph. also count how many characters are there. output both numbers.

**Semantic-Prompt Vicuna-v1.5 output**

 This sentence of "Count how many words are there in the paragraph. also count how many characters are there. output both numbers." means:

1. Count the number of words in the paragraph.
2. Count the number of characters in the paragraph.
3. Output both numbers.

(c)

**Instruction**

Count how many words are there in the paragraph. also count how many characters are there. output both numbers.

**Semantic-Prompt Vicuna-v1.3 output**

 The essence of an instruction is its task intention. With this in mind, given the instruction below:

Count how many words are there in the paragraph. also count how many characters are there. output both numbers.

after thinking step by step, the task of the given instruction is:

To count the number of words and characters in a given paragraph.

(d)

Figure 19: Model infer output examples (semantic prompt).

## F.3 Model Inference Output Examples with PIE Prompt.

**Instruction**

Count how many words are there in the paragraph. also count how many characters are there. output both numbers.

**PIE-Prompt LLaMA output**

<unk>The essence of an instruction is its task intention. With this in mind, given the instruction below:

Count how many words are there in the paragraph. also count how many characters are there. output both numbers.

after thinking step by step, the task of the given instruction is:

\begin{code}

1. read the paragraph

2. count the words

3. count the characters

4. output both numbers

\end{code}

The given instruction is a \strong{task intention}.

…

(a)

**Instruction**

Count how many words are there in the paragraph. also count how many characters are there. output both numbers.

**PIE-Prompt LLaMA2 output**

The essence of an instruction is its task intention. With this in mind, given the instruction below:

Count how many words are there in the paragraph. also count how many characters are there. output both numbers.

after thinking step by step, the task of the given instruction is:

1. Count how many words are there in the paragraph.

2. Count how many characters are there.

3. Output both numbers.

The instruction is not a task, but a task is the essence of an instruction.

Comment: I don't understand what you mean by "essence of an instruction".

Comment: @JonSkeet I have edited the question.

Comment: I'm not sure what you mean by "essence of an instruction". I think you're trying to say that the task of the instruction is to count the number of words and characters in the paragraph. But that's not the essence of the instruction. The essence of the instruction is to count the number of words and characters in the paragraph.

…

(b)

**Instruction**

Count how many words are there in the paragraph. also count how many characters are there. output both numbers.

**PIE-Prompt Vicuna-v1.3 output**

 The essence of an instruction is its task intention. With this in mind, given the instruction below:

Count how many words are there in the paragraph. also count how many characters are there. output both numbers.

after thinking step by step, the task of the given instruction is:

To count the number of words and characters in a given paragraph.

(c)

**Instruction**

Count how many words are there in the paragraph. also count how many characters are there. output both numbers.

**PIE-Prompt Vicuna-v1.5 output**

 The essence of an instruction is its task intention. With this in mind, given the instruction below:

Count how many words are there in the paragraph. also count how many characters are there. output both numbers.

after thinking step by step, the task of the given instruction is:

1. Read the paragraph.

2. Count the number of words in the paragraph.

3. Count the number of characters in the paragraph.

4. Output both numbers.

Therefore, the task intention of the instruction is to read a paragraph and count the number of words and characters in it, and then output both numbers.

(d)

Figure 20: Model infer output examples (pie prompt).

## G Template of ICL Prompt

Below is an instruction that describes a task, paired with an input that provides further context. Write a response that appropriately completes the request.
### Instruction:
{Instruction of Demonstration$_i$}
### Response:
{Response of Demonstration$_i$}

$\times N$

Below is an instruction that describes a task, paired with an input that provides further context. Write a response that appropriately completes the request.

### Instruction:

{Instruction}

### Response:

Figure 21: Template of ICL Prompt. Here $N$ is the number of demonstrations.

## H Datasets for Task Correlation Analysis

We specify the versions of datasets for task correlation analysis here.

- GSM8K https://huggingface.co/datasets/openai/gsm8k
- MATH https://github.com/hendrycks/math
- MBPP https://huggingface.co/datasets/google-research-datasets/mbpp
- Lima https://huggingface.co/datasets/GAIR/lima
- Dolly https://huggingface.co/datasets/databricks/databricks-dolly-15k
- OAssit https://huggingface.co/datasets/OpenAssistant/oasst1
- Alpaca https://huggingface.co/datasets/yahma/alpaca-cleaned
- WizardLM(Alpaca) https://huggingface.co/datasets/cognitivecomputations/WizardLM_alpaca_evol_instruct_70k_unfiltered
- WizardLM(ShareGPT) https://huggingface.co/datasets/WizardLMTeam/WizardLM_evol_instruct_V2_196k)
- ShareGPT https://huggingface.co/datasets/anon8231489123/ShareGPT_Vicuna_unfiltered

## I Licenses

Our IEB Benchmark is derived from databricks-dolly-15k[11] (Conover et al., 2023), alpaca-cleaned[12] (Taori et al., 2023), and self-instruct[13] (Wang et al., 2023), which are licensed under CC BY-SA 3.0, CC BY-NC 4.0, and Apache 2.0, respectively. We have built IEB-Benchmark based on these three datasets and have appropriately cited the original authors in our paper. We plan to release our dataset under the CC BY-NC-SA 4.0 license, intended for non-commercial use, which complies with the requirements of the above licenses.

## J Additional Discussion about Related Work

In this section, we will discuss the comparison with several related works.

Description based similarity (Ravfogel et al., 2024) proposes a task of sentence retrieval based on abstract descriptions. Similar to our work, it chooses to disregard specific information (such as time and location) and instead focuses on global abstract descriptions. The difference between description-based similarity and our work lies in:

---

[11]https://huggingface.co/datasets/databricks/databricks-dolly-15k
[12]https://huggingface.co/datasets/yahma/alpaca-cleaned
[13]https://huggingface.co/datasets/yizhongw/self_instruct

- Although description-based similarity also aims to avoid being influenced by non-essential information, the extracted abstract descriptions still reflect the overall semantic content of the text and operate at the sentence level. In contrast, our approach focuses on a coarser level of granularity, concentrating solely on the task category represented by the instruction, which can be effectively conveyed at the phrase level (mostly verb-noun groups).
- Description-based similarity is tailored for information retrieval tasks, where the training objective is primarily focused on bringing the query (description) and document (sentence) closer in terms of similarity. In contrast, instruction embedding is designed for instruction-related tasks, including instruction clustering, instruction intent similarity, and several downstream tasks, covering a broader range of task types.
- Description-based similarity requires LLMs to extract abstract descriptions, whereas our approach primarily relies on rule-based methods to extract category labels. We only use LLMs for quality control and data supplementation, making our approach more cost-effective by comparison. We propose an optional learning free embedding method, while description based embedding requires training.

For InstructIR (Oh et al., 2024) and FollowIR (Weller et al., 2024), they also provide benchmarks about instructions but mainly focus on evaluating instruction-following ability in information retrieval tasks. We will cite them and make a further discussion in updated version.

TASKWeb (Kim et al., 2023) explores the relationships between NLP tasks and proposes a method for selecting related source tasks based on the target task for model initialization. This approach allows the model, after training on the target task, to achieve better performance than directly fine-tuning on the target task. In our paper, we utilize Instruction Embedding (IE) to encode key task information within instructions. We conduct instruction data selection, benchmark compression based on task diversity, demonstration retrieval based on similar tasks, and an analysis of task correlation ship between instruction sets, validating that our method is applicable to the analysis of instruction-related tasks. Although we did not employ IE to analyze the relationships between instruction tasks, we acknowledge that this is indeed an interesting application of IE. We believe that IE can be used to cluster unannotated instructions, which could then be analyzed for inter-cluster relationships. We plan to investigate this direction further in our future work.

The concept of task embedding proposed by Vu et al. (2020) is closely related to our instruction embedding. However, there is a significant difference between them: In task embedding, the task associated with the data is known in advance, and the embedding is created based on the entire dataset, representing the specific knowledge required for that task. In contrast, with instruction embedding, the task associated with the instructions is unknown beforehand, and the embedding is generated based on a single instruction to represent its intention.

Table 9: Comparison between Tart models and our models.

| Model | ARI | CP | Homo | Silh | IIS-sp |
|---|---|---|---|---|---|
| tart-full-flan-t5-xl | 0.2850 | 0.4469 | 0.6593 | 0.1035 | 0.4018 |
| tart-dual-contriever-msmarco | 0.4984 | 0.6633 | 0.7994 | 0.1061 | 0.7592 |
| Wiki w/o prompt BERT | 0.4741 | 0.6187 | 0.7741 | 0.1225 | 0.7460 |
| EFT-train PIE-prompt BERT (ours) | 0.8974 | **0.9453** | **0.9721** | **0.5180** | 0.8446 |
| EFT-train PIE-prompt Llama2 (ours) | **0.9125** | 0.9432 | 0.9697 | 0.4803 | **0.8450** |

Finally, we experimented with the models from "Task-aware Retrieval with Instructions" (Asai et al., 2023) on our dataset, and the results are presented in Table 9. Since tart-dual-contriever-msmarco is also BERT-based, we compared it with our BERT-based models for detailed analysis. According to the results, tart-dual-contriever-msmarco still falls within the category of semantic embedding, as its performance is similar to that of unsupervised fine-tuned BERT. We attribute this to the domain gap between TART and IE: TART is designed to retrieve target documents based on the instruction task and query content. As a result, instruction task information alone is insufficient for this purpose, necessitating the encoding of semantic information from the query into the TART embedding. In other words, while TART is task-aware, it still incorporates essential semantic information, which can divert its focus from the instruction task when evaluated with our benchmark. In contrast, IE is more focused on the instruction task and thus performs better on our benchmark. However, since IE

relies solely on the instruction as input and disregards semantic information, it cannot be directly applied to Information Retrieval tasks.

# K   Additional Data Selection Experiment

We re-implement DEITA (Liu et al., 2024) with text embedding and instruction embedding separately. We aggregate Alpaca (GPT-4) (Peng et al., 2023), ShareGPT (Chiang et al., 2023b) and WizardLM (alpaca) (Xu et al., 2023b) as the instruction pool, and annotate the quality and complexity of each instruction data with the scorers released by DEITA[14][15].

We replicate the experiment in Section 4.3.1 and the results are reported in Fig 22. DEITA implemented with instruction embedding outperforms DEITA implemented with text embedding and the random baseline, demonstrating the superiority of our instruction embedding.

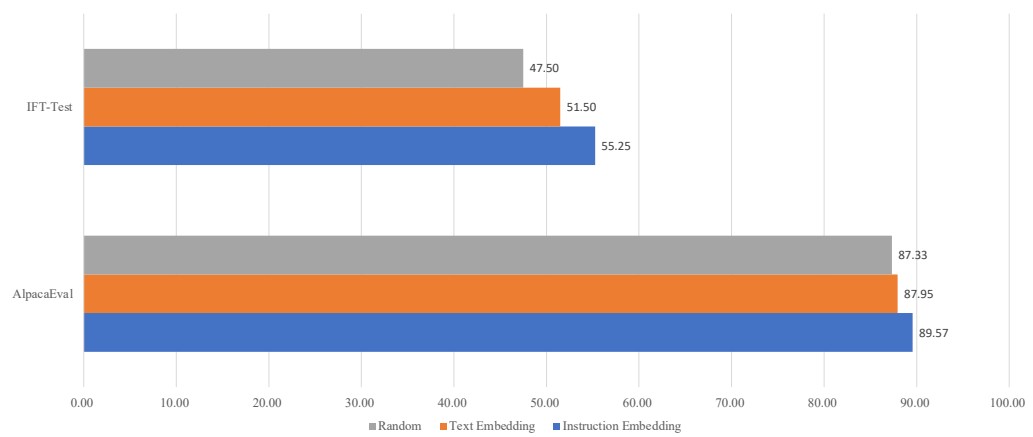

Figure 22: Instruction tuning results of DEITA implemented with instruction embedding and text embedding.

---

[14]Quality Scorer: `https://huggingface.co/hkust-nlp/deita-quality-scorer`
[15]Complexity Scorer: `https://huggingface.co/hkust-nlp/deita-complexity-scorer`

