# OpenReview forum: "Instruction Embedding: Latent Representations of Instructions Towards Task Identification"
_NeurIPS.cc/2024/Datasets_and_Benchmarks_Track — NeurIPS 2024 Track Datasets and Benchmarks Poster_

### Official Review · Reviewer_4oB5 · 2024-07-23

**Rating:** 6
**Confidence:** 3
**Correctness:** Yes.
**Clarity:** Yes.

**Review:**

### Strengths
1. The paper is well-written and easy to follow.
2. The data and code are available via supplementary materials.
3. The task of instruction embedding is new and practical and the introduced benchmark is the first work in this area as far as I know.
4. The proposed baseline method is simple yet effective.

### Weaknesses
1. GPT-4 is employed for data synthesis and filtering, which may still result in noise.
2. There are overlapping tasks between the training and testing sets. It remains uncertain whether a model that performs well on the proposed benchmark will still be effective in real-world applications, particularly for tasks that are out-of-distribution.
3. The foundational models utilized for the proposed PIE method are large and inefficient. In practical applications, using such extensive models for data filtering leads to significant computational overhead.

**Strengths:**

See **Strengths** in #Review.

**Additional Feedback:**

n/a

**Documentation:**

Yes.

**Ethics:**

No.

**Limitations:**

No. It is recommended to discuss the limitations related to data noise, model transferability, and computational overhead.

**Opportunities For Improvement:**

See **Weaknesses** in #Review.

**Relation To Prior Work:**

Yes.

**Summary And Contributions:**

The paper introduces a new task of instruction embedding which aims to identify tasks among instructional samples. The task benefits downstream tasks, e.g., data selection, and demonstration retrieval. The authors propose an instruction embedding benchmark (IEB) with 47k samples in total. The benchmark can be used for both training and testing. Furthermore, they propose a baseline method, namely, prompt-based instruction embedding (PIE) with both training-free and fine-tuning settings. The effectiveness of PIE has been verified on the proposed IEB.

---

> ### Author Rebuttal · Authors · 2024-08-21
>
> Thank you for the valuable and detailed review! We understand your concerns, which are also very important to us. Below we clarify:
>
> >Noise from GPT-4
>
> For filtering, we manually verified the correctness of the data, with the vast majority of samples being accurately assessed (approximately 95\%). Data synthesis constitutes only a small portion, and its correctness has also been validated through quality control described in Section 2.3. As is well known, manual filtering and synthesis are costly, so leveraging GPT-4 offers a cost-effective yet efficient alternative. The manual validation results indicate that 93\% of the sample categories are accurate. We acknowledge that using GPT-4 may introduce some noise, but we believe that the results are already quite acceptable.
>
> >Overlapping issue
>
> Generalization is indeed a significant issue, as it relates to the model's effectiveness in real-world applications. However, our experiments are designed to ensure generalization, as clarified below:
> 1. For EFT-train and EFT-test, we ensured that there are no overlapping tasks between these two datasets during partitioning. Therefore, the main experimental results presented in Section 4.2 do not raise any generalization concerns.
> 2. For IFT-train and IFT-test, there is some overlap in tasks between these datasets. This was done to ensure that both the instruction fine-tuning training and test sets cover sufficient task categories. At the same time, we have carefully considered the associated generalization concerns. To address these, we used AlpacaEval as a held-out test set. Results from AlpacaEval, along with those from IFT-test, are both reported in our paper, as shown in Figure 4.
>
> >Model efficiency
>
> Efficiency is a critical factor in real-world applications. Initially, we selected Llama2-7b as the base model for PIE to maintain consistency with the base model used in the respective downstream tasks. Recognizing the practical importance of efficiency, we also performed EFT-related training and evaluation using BERT-base as the base model, with the results presented in Table 3.
> As depicted in Table 3, when not fine-tuned on EFT-train set, Llama2 exhibits superior instruction-following capabilities compared to BERT, in clustering and IIS tests with the PIE-prompt. However, after supervised fine-tuning, this performance gap narrows significantly, with BERT's performance in clustering and IIS tests becoming comparable to, or in some cases surpassing, that of Llama2. This suggests that BERT-based instruction embeddings possess a quality comparable to that of Llama2-based instruction embeddings which is efficient and practical.
>
> >Limitations
>
> Thanks for your reminding, we add the limitations here:
> 1. Not relying entirely on manual labeling or checking means that not all the data is of high quality. The manual validation results indicate that 93\% of the sample categories are accurate, which means there is still a small portion of data that may contain noise.
> 2. We have not addressed the case of multi-step instructions, where several serialized tasks are included within the same instruction. Since no such cases have been manually identified in the chosen dataset, we haven't separately handled or supplemented these samples.
> 3. The three popular instruction datasets we selected are all single-turn, so the benchmark does not include multi-turn samples.

---

> ### Author Response · Authors · 2024-08-27
> **Kindly Reminder**
>
> Thank you again for your constructive suggestions! We hope our response adequately addresses your concerns. Please let us know if you have any further questions, as we are happy to continue the discussion.

---

### Official Review · Reviewer_35T1 · 2024-07-24

**Rating:** 5
**Confidence:** 4
**Correctness:** Yes
**Clarity:** Yes

**Review:**

Pros and Cons

Pros:
- The idea is interesting and provides a new benchmark for retrieval models
- There is good potential downstream implications for other tasks that they evaluate on like data selection and ICL examples
- The results show improvements in many cases

Cons:
- I think I'm most unsure at the results section: what about a random baseline? Are these results significant, e.g. with error bars? The tiny benchmark looks especially sensitive to randomness but that isn't considered.
- Although the motivation of the paper seems to be to get away from lexical-semantic information, it seems like the dataset is created using that? E.g. based on whether it is a yes/no question or uses a verb+noun or even just "is about math". I think this idea has potential but the dataset seems to be haphazardly created: sometimes the distinction is based on domain, sometimes on standard NLP-like task. Perhaps I'm misunderstanding, but the process seems unclear for generating the dataset.
- The paper fails to cite the most relevant paper to it, "Description-Based Text Similarity" which focuses on a broader version of this claim that works beyond just instruction-based text.  I would like to see some comparison between the works, at least in a discussion section.
- The baselines are a bit confusing: what model is the "text embedding" in orange in Figure 4? It seems perhaps a Wiki CSE trained model and is that a fair comparison in terms of data for training, etc.?

Overall I think the motivation and downstream implication could be very exciting but there are some unresolved issues that make it so that I am not convinced of the thoroughness of the evaluation

**Strengths:**

- The motivation is strong and the examples in the paper are nice!
- The visualizations are especially well done, very pretty!
- The chosen downstream implications seem like a good evaluation for this, since the regular evaluation seems basically solved (?) with 90+ ARI, etc.

**Additional Feedback:**

N/A

**Documentation:**

N/A

**Opportunities For Improvement:**

- Clarifying data creation process
- Making the results section more robust in Section 4.

**Relation To Prior Work:**

Mostly. Some references:
- InstructIR: A Benchmark for Instruction Following of Information Retrieval Models
- FollowIR: Evaluating and Teaching Information Retrieval Models to Follow Instructions
- Description-Based Text Similarity

**Summary And Contributions:**

This paper proposes to do retrieval via similarity to an instruction meaning rather than to the lexical/semantic meaning. They also show that doing an expansion with an LM asking it to produce task meaning words makes it easier to do the search (PIE for prompt-based instruction embedding). They finetune embedding models using CSE and their data and show improved results with the PIE prompt and with fine-tuning.

---

> ### Author Rebuttal · Authors · 2024-08-21
>
> >Comparison with Description-Based Text Similarity and other related work
>
> Thanks a lot for your reminder! We will discuss the comparison with "Description-Based Text Similarity" [1] and cite it in the updated version: Description based similarity proposes a task of sentence retrieval based on abstract descriptions. Similar to our work, it chooses to disregard specific information (such as time and location) and instead focuses on global abstract descriptions. The difference between description-based similarity and our work lies in：
> 1. Although description-based similarity also aims to avoid being influenced by non-essential information, the extracted abstract descriptions still reflect the overall semantic content of the text and operate at the sentence level. In contrast, our approach focuses on a coarser level of granularity, concentrating solely on the task category represented by the instruction, which can be effectively conveyed at the phrase level (mostly verb-noun groups).
> 2. Description-based similarity is tailored for information retrieval tasks, where the training objective is primarily focused on bringing the query (description) and document (sentence) closer in terms of similarity. In contrast, instruction embedding is designed for instruction-related tasks, including instruction clustering, instruction intent similarity, and several downstream tasks, covering a broader range of task types.
> 3. Description-based similarity requires LLMs to extract abstract descriptions, whereas our approach primarily relies on rule-based methods to extract category labels. We only use LLMs for quality control and data supplementation, making our approach more cost-effective by comparison.
> 4. We propose an optional learning free embedding method, while description based embedding requires training.
>
> For InstructIR [2] and FollowIR [3], they also provide benchmarks about instructions but mainly focus on evaluating instruction-following ability in information retrieval tasks. We will cite them and make a further discussion in updated version.
>
> [1] Ravfogel, Shauli, et al. Description-Based Text Similarity. COLM 2024.
>
> [2] Oh, Hanseok, et al. INSTRUCTIR: A Benchmark for Instruction Following of Information Retrieval Models. arXiv:2402.14334.
>
> [3] Weller, Orion, et al. FollowIR: Evaluating and Teaching Information Retrieval Models to Follow Instructions. arXiv:2403.15246.
>
> >Details about text embedding
>
> As mentioned in the first paragraph of Section 4.3, we fine-tuned Llama2-7b on Wiki. Our core goal, as stated in this section, is to demonstrate that instruction embedding is more effective for instruction-related tasks than text embedding. Therefore, we have chosen the strongest model for each concept. Both of them are trained on Llama2-7b, so the model size is consistent. Given that the Wiki dataset is much larger than ours, the text embedding results will naturally be stronger, ensuring that there is no unfair comparison.

---

> ### Author Rebuttal · Authors · 2024-08-21
>
> Thank you for the valuable and detailed review! We understand your concerns, which are also very important to us. Below we clarify:
>
> >Random baseline
>
> Thank you for your suggestion! We agree that random selection is an important baseline, and we did not include it in the paper due to limit space and its natural low performance. Now we supplement the relevant experiments. For Data selection for instruction tuning, we conduct 5 random selections, and for demonstrations retrieval and tiny benchmark selection, we conducted 10 selections. We calculated the mean and variance of the results, which are shown below:
>
> Data selection for instruction tuning:
> |Method|IFT-test|AlpacaEval|
> |---|---|---|
> |Instruction Embedding|**25.50**|**13.77**|
> |Random|21.00 (2.51)|7.70 (3.10)|
>
> Demonstrations retrieval for in-context learning:
> | Method            | IFT (Instruction / Random)  | AlpacaEval (Instruction / Random) |
> |-------------------|----------------------------|----------------------------------|
> | Vicuna            | 34.92 / **36.00 (2.85)**   | **38.48** / 33.58 (1.07)         |
> | Llama2-chat       | **39.00** / 38.59 (2.45)   | **42.00** / 36.45 (1.24)         |
> | Mistral           | 38.50 / **41.08 (3.66)**   | **56.41** / 49.17 (1.58)         |
> | LongChat          | **31.00** / 28.72 (2.22)   | **32.90** / 26.68 (1.34)         |
>
>  Tiny benchmark:
> | Method            | 10 samples (Instruction / Random)  | 50 samples (Instruction / Random) |  100 samples (Instruction / Random) |
> |-------------------|----------------------------|----------------------------------|----------------------------------|
> | Vicuna            | **1.02** / 10.41 (9.84)   | **4.08** / 6.48 (5.00)         | **1.53** / 5.83(4.07)         |
> | Llama2-chat       | **11.59** / 18.00 (12.3)   | 8.40 / **7.76 (5.90)**         | **2.03** / 3.68 (1.55)         |
> | Mistral           | **7.05** / 9.26 (8.37)   | **2.94** / 5.38 (4.07)         | **2.23** / 4.77 (3.64)         |
> | LongChat          | 17.08 / **14.95 (10.07)**  | **0.64** / 6.82 (3.43)         | **1.70** / 4.81 (2.86)         |
>
> From the results, we can see that random selection indeed performs worse (lower mean values) compared to the instruction embedding-based method. It also has a significantly higher standard deviation, indicating poor stability, making it unsuitable for practical use.
>
>
> >Clarification about data creation process
>
> We want to clarify that we are not getting rid of lexical semantics, and getting rid of semantic information does not mean abandoning all token semantic information. What we aim to emphasize is avoiding the influence of irrelevant content in the text and instead focusing more on the task (key words) itself. For example, "writing an article" is a task, but the specific topic ("writing an article about XXX") of the article is not important. We acknowledge that the criteria for determining categories are not so uniform or singular, as we want to encompass as much data as possible from the real instruction dataset. However, the forms and categories of instructions are too diverse, making it difficult to label categories based on a single parsing tag using our rule-based approach (i.e. parsing). Otherwise, too many samples will be discarded, resulting in an insufficiently rich and robust dataset. The different parsing tags and annotations that we show in Table 1 revolve around the core idea: distinguishing the category of instructions based solely on key words. Therefore, different numerical values in math problems and different knowledge in questions will not affect our categorization, but for example, we will roughly classify questions based on the category of interrogative pronouns.

---

> > ### Comment · Reviewer_35T1 · 2024-08-21
> >
> > Thanks for the response. This answers most of my concerns about the random method although I am having some trouble in parsing the table (I know it's hard in markdown). E.g. can you help me parse the line "1.02 / 10.41 (9.84)" for Vicuna on Tiny: is that the instruction models got 1.02 and the Random got 10.41 (with 0.84 std)? Or vice versa?
> >
> > Otherwise I think the only unanswered concern I have was about the variance for each of the text-embedding and the instruction-embedding models. Is that in these tables also and I missed it?
> >
> > Thank you also for the detailed comparison to the related work and the explanations, I understand and they are resolved now.

---

> > > ### Author Response · Authors · 2024-08-27
> > > **Kindly Reminder**
> > >
> > > Thank you again for your constructive suggestions! We hope our response adequately addresses your concerns. Please let us know if you have any further questions, as we are happy to continue the discussion.

---

> > ### Author Rebuttal · Authors · 2024-08-22
> >
> > Thanks for your response! Sorry for the misunderstanding. The previously reported experimental results for instruction embedding and text embedding were based on a single run because we initially considered that data selection based on clustering would be more deterministic compared to random selection. However, thanks to your feedback, we realized that the randomness in the initial point selection of the K-means clustering method can indeed affect the clustering results to some extent. Therefore, we repeated multiple experiments (5 for data selection for instruction tuning and 10 for tiny benchmark similarly to random selection) and calculated the mean value and standard deviation. For the demonstrations retrieval task, since the demos are selected directly based on distance, the process is deterministic, so multiple runs are not necessary. Below, we present the revised experimental results for Data Selection for Instruction Tuning and Tiny Benchmark (\%), with the standard deviation values indicated in bracket:
> >
> > |Method|IFT-test|AlpacaEval|
> > |---|---|---|
> > |Instruction Embedding|**23.75 (1.81)**|**10.96 (1.98)**|
> > |Text Embedding|19.00 (3.32)|6.74 (1.47)|
> > |Random|21.00 (2.51)|7.70 (3.10)|
> >
> > | Model            | 10 samples (Instruction / Text / Random)  | 50 samples (Instruction / Text / Random) |  100 samples (Instruction / Text / Random) |
> > |-------------------|----------------------------|----------------------------------|----------------------------------|
> > | Vicuna            | **6.61 (6.43)** / 17.02 (12.81) / 10.41 (9.84)   | **3.84 (2.99)** / 8.22 (4.19) / 6.48 (5.00)         | **1.73 (1.15)** / 3.43 (3.30) / 5.83(4.07)         |
> > | Llama2-chat       | **12.90 (8.68)** / 21.60(8.94) / 18.00 (12.3)   | 4.98 (4.03) / **4.82 (3.71)** / 7.76 (5.90)         | **3.42 (2.24)** / 3.44 (2.40) / 3.68 (1.55)         |
> > | Mistral           | 8.90(5.46) / **7.59 (4.84)** /9.26 (8.37)   | **1.99 (1.33)** / 4.21 (2.73) /5.38 (4.07)         | **1.85 (1.07)** / 3.20 (2.24) /4.77 (3.64)         |
> > | LongChat          | **11.31 (8.01)** / 14.34 (9.21) / 14.95 (10.07)   | **3.80 (2.31)** / 5.41 (3.35) / 6.82 (3.43)         | **4.17 (2.88)** / 3.97 (2.06) / 4.81 (2.86)         |
> >
> > The results show that instruction embedding outperforms both text embedding and random selection, with the smallest variance. This is because we found that the intersection of results from multiple embedding-based selections is around 50\%, while the overlap in random selections is very low. The variance of instruction embedding is smaller than that of text embedding because its space (as seen in the visualization results) is more dispersed, leading to more stable clustering outcomes. We will update the experimental results for 4.3 section in the paper.

---

> ### Comment · Reviewer_35T1 · 2024-08-27
>
> Thanks for the results!
>
> This is what I thought was the case, there's a lot of variance.  I don't think any of the instruction retrieval results are statistically significant compared to random (I plugged the Instruction Embedding vs Random from IFT-test into a t-test calculator and came back 0.08 p-value and most of the tiny benchmark is also much less significant).
>
> The previous table about the ICL demonstrations also shows random being statistically similar to the instruction embedding for IFT. I think the only dataset that is statistically significant is AlpacaEval ICL.  I think that leaves many questions: why does it only convincingly help Alpaca but not any of the others (and perhaps after figuring that out, getting more datasets like AlpacaEval for ICL experiments).
>
> Overall, I do think the paper is interesting but it doesn't seem there are convincing results for downstream implications - and I was already unsure of the usefulness of the main task itself. I leave my score unchanged, although I do thank the authors for providing these results to clear up the confusion.

---

> > ### Author Rebuttal · Authors · 2024-08-30
> >
> > Thanks for your response! We understand your concerns about the significance of the experimental results. We have conducted additional experiments, and we would like to clarify the following details to address your concerns, highlighting the significant performance of our method in downstream tasks.
> >
> > 1.Data Selection for Instruction Tuning
> >
> > As previously mentioned, the variance in the experimental results mainly stems from the randomness of the clustering algorithm's initial points. For data selection for instruction tuning, we reproduce the SOTA method, DEITA [1], as its data selection approach for diversity objective relies solely on distance calculations and does not require clustering, making it deterministic. Moreover, directly applying instruction embeddings to a SOTA selection method, rather than naively selecting after clustering, better demonstrates its practical value in real-world scenarios. We apply distance calculations using both instruction embedding and text embedding, and compare the results with random selection. DEITA Diversity Only is the method that considers only diversity, whereas DEITA represents the full multi-objective compression approach in [1]. Similarly to DEITA, we combine the datasets WizardLM (Alpaca), WizardLM (ShareGPT), Alpaca (GPT-4), Dolly, and ShareGPT, and compress them down to 6,000 samples. We train the compressed data on Llama3-8B and test it on AlpacaEval (with GPT-3.5 Turbo and GPT-4 as references). The results for 3 runs are as follows:
> >
> > |Method|AlpacaEval (gpt-3.5-turbo-1106)|AlpacaEval (gpt-4)|
> > |-------------------|----------------------------|----------------------------|
> > |DEITA Diversity Only (Instruction Embedding)|39.44* (0.38)|15.33* (0.63)|
> > |DEITA Diversity Only (Text Embedding)|35.62 (0.82)|10.99 (0.86)|
> > |DEITA (Instruction Embedding)|**46.77\*\* (0.66)**|**19.77\*\* (0.41)**|
> > |DEITA (Text Embedding)|46.08(0.32)|18.84 (0.17)|
> > |Random|37.04 (1.34)|13.54 (0.87)|
> >
> > The experimental results indicate that instruction embedding achieved significantly superior outcomes. Since the selected data is deterministic, the variance only stems from the random seeds used during training. * denotes P-value < 0.05, and ** denotes < 0.01.
> >
> > 2.Demonstrations Retrieval
> >
> > Besides AlpacaEval, we supplement the experimental results with the MT-bench [2], which shows that our method consistently outperforms random selection and text embedding. AlpacaEval and MT-bench are currently the two most prominent benchmarks for evaluating instruction-following capabilities. The outstanding performance of our method on these benchmarks indicates its significance in the Demonstrations Retrieval task. We also conduct a manual check and analysis of the selected examples, confirming that the distance measure based on instruction embedding can effectively identify demonstrations that are very similar to the input instructions.
> >
> > | Model            | Instruction Embedding | Text Embedding| Random |
> > |-------------------|----------------------------|----------------------------|----------------------------|
> > | Vicuna            | **5.49 (-)** | 5.01 (-) | 5.42 (0.15) |
> > | Llama2-chat       | **5.41 (-)** | 4.93 (-) | 4.98 (0.08) |
> > | Mistral           | **6.36 (-)** | 5.95 (-) | 6.24 (0.14) |
> > | LongChat          | **5.35 (-)** | 4.19 (-) | 4.74 (0.23) |

---

> > ### Author Rebuttal · Authors · 2024-08-30
> >
> > 3.Tiny Benchmark
> >
> > Test set compression does indeed introduce considerable randomness, especially considering that the evaluation of instructions is essentially a binary classification task, which results in high variance. Thus we observe that in Anchor point [3],  an important test set compression method, they conducted over 100 randomized runs using , so we follow it and conduct 100 experiments as well. Given that our sample size exceeds 30, we opt to perform Z-test, and the results are as follows:
> >
> > | Model            | 10 samples (Instruction / Text / Random)  | 50 samples (Instruction / Text / Random) |  100 samples (Instruction / Text / Random) |
> > |-------------------|----------------------------|----------------------------------|----------------------------------|
> > | Vicuna            | **8.92\* (3.97)** / 13.22 	(6.10) / 11.53 	(10.13)  | **3.76\*\*\* 	(2.88)**  / 8.56 	(5.17)  / 5.88 	(4.50) | **3.43\*\* (2.28)** / 3.61 (2.35)  / 4.61 	(3.15)  |
> > | Llama2-chat       |18.40 (5.81) / 33.50 (6.11) / **13.46 (10.24)**  |6.89 	(4.28) / **5.35 	(4.08)** / 5.68 	(4.24) | **3.17\* 	(2.28)**  / 3.34 	(2.39)  / 3.97 	(2.82)  |
> > | Mistral           | 7.92\*\*\* 	(3.31) / **2.98 (0.41)** / 10.94 	(8.47) | **4.27\*\* 	(3.04)**  / 5.05 	(3.44) / 5.67 	(3.84) | **2.14\*\*\*	(1.86)**  /  3.29 	(2.40)  / 3.35 	(2.28)  |
> > | LongChat          |**7.76\*\* 	(4.69)**  / 28.82 	(4.63) /  12.07 	(9.90)  |**4.69\*\* 	(3.60)**  /4.74 (3.68) / 6.11 	(4.07) | 3.70 	(2.42)  / **3.47 	(2.42)**  / 4.22 	(3.13)  |
> >
> > The experimental results indicate that with more repetitions, the results become more stable, and instruction embedding demonstrates more significant outcomes. * denotes P-value < 0.05, ** denotes < 0.01, and *** denotes < 0.001. It is important to note that, similar to the Data Selection for Instruction Tuning task, clustering-based selection is just a naive method for quick evaluation, and the improvement over random selection would be limited. In this paper, our primary focus is to introduce a novel approach for latent representations of  instructions, rather than designing detailed downstream task schemes. To the best of our knowledge, the currently published test set compression methods do not utilize text embeddings, which makes it difficult to directly apply to SOTA test set compression methods.
> >
> > In summary, we are very grateful for your thorough feedback, which has helped make our experimental results more reliable and robust. We believe that the current experimental setup and results demonstrate the effectiveness of proposed instruction embedding while also highlighting its potential for deeper applications in the future.
> >
> > [1] Liu, Wei, et al.  What Makes Good Data for Alignment? A Comprehensive Study of Automatic Data Selection in Instruction Tuning. ICLR 2024.
> >
> > [2] Zheng, Lianmin, et al. Judging llm-as-a-judge with mt-bench and chatbot arena. Neurips 2024.
> >
> > [3] Vivek, Rajan, et al. Anchor Points: Benchmarking Models with Much Fewer Examples. EACL 2024.

---

### Official Review · Reviewer_uG4q · 2024-07-28
**Offical Review by Reviewer uG4q**

**Rating:** 7
**Confidence:** 3
**Correctness:** Yes, the claims are correct to the be…
**Clarity:** Yes. The writing is good. The paper i…

**Review:**

Pros
1. This paper proposes an interesting task to identify the task behind each instruction.
2. Detailed analyses of the built benchmark are provided.
3. The study on instruction embedding is comprehensive, including the new benchmark, task, and method.
4. Code is available during the reviewing phase.

Cons
1. Figure 1, instead of comparing absolute cosine similarity scores, I would suggest reporting a relative score compared to the average similarity score. The absolute scores may be affected by the anisotropy issue of pre-trained language models, resulting in a biased score distribution. Directly comparing different models with different similarity score distributions by their absolute cosine similarity could be inaccurate.
2. In lines 125 - 128, the authors filter out those categories with fewer than 10 samples. Would this lead to a popularity bias, i.e., the benchmark favors models that can well identify popular instructions?
3. In Table 2, I would suggest the authors describe the meanings of EFT and IFT in more detail.
4. The related work is too concise. Could the background of the described task be properly introduced using only 9 papers? In addition, I would suggest the authors add more related works about the instruction tuning of LLMs.
5. No limitations or risks are discussed in this paper.
6. The authors use existing datasets to construct the benchmark. However, the licenses of existing datasets are not discussed.

**Strengths:**

1. This paper proposes an interesting task to identify the task behind each instruction.
2. Detailed analyses of the built benchmark are provided.
3. The study on instruction embedding is comprehensive, including the new benchmark, task, and method.
4. Code is available during the reviewing phase.

**Additional Feedback:**

N/A

**Documentation:**

Yes.

**Limitations:**

No. I would suggest explicitly discussing the limitations and risks in this paper.

**Opportunities For Improvement:**

Please refer to "Cons" in "Review" for more details

**Relation To Prior Work:**

The related work is too concise. I recommend finding additional related works and providing a more comprehensive discussion. Additionally, consider adding a new section specifically focused on the instruction tuning of LLMs.

**Summary And Contributions:**

The paper defines a task that evaluates how well embedding models can identify the underlying tasks based on given instructions. Based on existing instruction datasets, the authors carefully define several categories and curate the raw datasets into a benchmark. The authors further introduce a method named PIE for the instruction embedding task.

---

> ### Author Rebuttal · Authors · 2024-08-21
>
> >Related works about the instruction tuning
>
> Thanks for your advice! In fact, we have already reviewed related works about instruction tuning of LLMs, but we removed this content due to page limitations. We will add it back into the paper. The related works about instruction tuning of LLMs are shown below:
>
> **Instruction Tuning**
>
> While large language models (LLMs) have demonstrated remarkable capabilities in natural language processing, they still struggle to follow human instructions appropriately due to a mismatch between their training objectives and user intent. LLMs are initially trained to predict the next token in a sequence, whereas users expect them to follow instructions in a helpful and safe manner [1]. To address this objective mismatch, instruction tuning has been proposed. Early approaches to instruction tuning focused on fine-tuning LLMs with large amounts of instruction data [2,3,4,5] or aligning LLMs with users through reinforcement learning from human feedback (RLHF), which is collected from millions of interactions with human annotators [1]. The LIMA framework argues that existing alignment methods require substantial computational resources and specialized data to achieve ChatGPT-level performance, emphasizing the importance of data quality and diversity over sheer quantity [6]. Building on LIMA’s insights, recent efforts have focused on compressing instruction datasets. For instance, ALPAGASUS [7] uses ChatGPT to filter out low-quality data, Humpback [8] selects high-quality examples through an iterative self-curation process, and DIVERSEEVOL [9] iteratively samples training data using the current embedding space to preserve diversity within the subset. However, previous efforts have not explicitly maintained task diversity within the training subset while reducing data quantity due to the inability to explicitly identify instruction tasks.
>
> Referecnes
>
> [1] Ouyang, Long, Jeffrey Wu, Xu Jiang, Diogo Almeida, Carroll Wainwright, Pamela Mishkin, Chong Zhang et al. "Training language models to follow instructions with human feedback." Advances in neural information processing systems 35 (2022): 27730-27744.
>
> [2] Wei, Jason, Maarten Bosma, Vincent Y. Zhao, Kelvin Guu, Adams Wei Yu, Brian Lester, Nan Du, Andrew M. Dai, and Quoc V. Le. "Finetuned language models are zero-shot learners." arXiv preprint arXiv:2109.01652 (2021).
>
> [3] Wang, Yizhong, Swaroop Mishra, Pegah Alipoormolabashi, Yeganeh Kordi, Amirreza Mirzaei, Atharva Naik, Arjun Ashok et al. "Super-NaturalInstructions: Generalization via Declarative Instructions on 1600+ NLP Tasks." In Proceedings of the 2022 Conference on Empirical Methods in Natural Language Processing, pp. 5085-5109. 2022.
>
> [4] Wang, Yizhong, Yeganeh Kordi, Swaroop Mishra, Alisa Liu, Noah A. Smith, Daniel Khashabi, and Hannaneh Hajishirzi. "Self-Instruct: Aligning Language Models with Self-Generated Instructions." In Proceedings of the 61st Annual Meeting of the Association for Computational Linguistics (Volume 1: Long Papers), pp. 13484-13508. 2023.
>
> [5] Taori, Rohan, Ishaan Gulrajani, Tianyi Zhang, Yann Dubois, Xuechen Li, Carlos Guestrin, Percy Liang, and Tatsunori B. Hashimoto. "Alpaca: A strong, replicable instruction-following model." Stanford Center for Research on Foundation Models. https://crfm. stanford. edu/2023/03/13/alpaca. html 3, no. 6 (2023): 7.
>
> [6] Zhou, Chunting, Pengfei Liu, Puxin Xu, Srinivasan Iyer, Jiao Sun, Yuning Mao, Xuezhe Ma et al. "Lima: Less is more for alignment." Advances in Neural Information Processing Systems 36 (2024).
>
> [7] Chen, Lichang, Shiyang Li, Jun Yan, Hai Wang, Kalpa Gunaratna, Vikas Yadav, Zheng Tang et al. "Alpagasus: Training a better alpaca with fewer data." arXiv preprint arXiv:2307.08701 (2023).
>
> [8] Li, Xian, Ping Yu, Chunting Zhou, Timo Schick, Luke Zettlemoyer, Omer Levy, Jason Weston, and Mike Lewis. "Self-alignment with instruction backtranslation." arXiv preprint arXiv:2308.06259 (2023).
>
> [9] Wu, Shengguang, Keming Lu, Benfeng Xu, Junyang Lin, Qi Su, and Chang Zhou. "Self-evolved diverse data sampling for efficient instruction tuning." arXiv preprint arXiv:2311.08182 (2023).
>
> >Limitations
>
> Thanks for your reminding, we add the limitations here:
> 1. Not relying entirely on manual labeling or checking means that not all the data is of high quality. The manual validation results indicate that 93\% of the sample categories are accurate, which means there is still a small portion of data that may contain noise.
> 2. We have not addressed the case of multi-step instructions, where several serialized tasks are included within the same instruction. Since no such cases have been manually identified in the chosen dataset, we haven't separately handled or supplemented these samples.
> 3. The three popular instruction datasets we selected are all single-turn, so the benchmark does not include multi-turn samples.
>
> >Licenses
>
> Our IEB Benchmark is derived from databricks-dolly-15k [1], alpaca-cleaned [2], and self-instruct [3], which are licensed under CC BY-SA 3.0, CC BY-NC 4.0, and Apache 2.0, respectively. We have built IEB-Benchmark based on these three datasets and have appropriately cited the original authors in our paper. We plan to release our dataset under the CC BY-NC-SA 4.0 license, intended for non-commercial use, which complies with the requirements of the above licenses.
> We will add a detailed discussion about the dataset's licensing in the revised version of the paper.
>
> [1] Free Dolly: Introducing the World's First Truly Open Instruction-Tuned LLM https://huggingface.co/datasets/databricks/databricks-dolly-15k
>
> [2] Stanford Alpaca: An Instruction-following LLaMA model https://huggingface.co/datasets/yahma/alpaca-cleaned
>
> [3] Self-Instruct: Aligning Language Model with Self Generated Instructions https://huggingface.co/datasets/yizhongw/self\_instruct

---

> ### Author Rebuttal · Authors · 2024-08-21
>
> Thank you for the valuable and detailed review! We understand your concerns, which are also very important to us. Below we clarify:
>
> >Relative score of figure1
>
> This is a good suggestion. According to your suggestion, we first calculated the average similarity score for text embeddings and instruction embeddings.
> For text embeddings, the average similarity score between samples is 0.4167.
> For instruction embeddings, the average similarity score between samples is 0.0452.
> Furthermore, we report the relative similarity score of two pairs of samples in Figure 1 with respect to their respective average similarity scores.
> For sample 1, the relative similarity for text embeddings is 2.38, while for instruction embeddings it is -0.56.
> For sample 2, the relative similarity for text embeddings is 0.777, while for instruction embeddings it is 18.334.
>
> >Filtering categories with fewer than 10 samples
>
> Thank you for your question! As mentioned in the paper, we found that task categories with fewer samples indeed had very poor quality (with over 60\% being noise samples). In fact, as shown in Figure 9 of the Appendix about Benchmark, which displays the distribution of the number of instructions per category, many categories had a large number of samples. Meanwhile, considering that the overall dataset contains nearly 50,000 samples, a category with only 10 samples is indeed relatively unpopular. For categories with more than 100 samples, we have already randomly retained only 100 to alleviate popularity bias. As you have pointed out, this direct filtering approach does result in the exclusion of some very rare categories. However, given the importance of quality control for the benchmark, we had to make the difficult decision to omit this part.
>
> >Details of EFT and IFT
>
> As indicated in the caption of Table 2, EFT denotes embedding fine-tuning, while IFT represents instruction fine-tuning. We divided the full IEB dataset into two subsets: EFT and IFT, each consisting of a training set and a test set. The EFT subset is designed to facilitate models in generating high-quality latent representations of instructions through embedding fine-tuning, which involves a supervised contrastive learning process based on our task labels (details on the embedding fine-tuning process can be found in Sections 3.2, 4.1, and 4.2). The IFT subset is constructed to evaluate the effectiveness of our instruction embeddings in downstream tasks, such as Data Selection for Instruction Tuning and Demonstration Retrieval (details available in Sections 4.3.1 and 4.3.2).

---

> ### Comment · Reviewer_uG4q · 2024-08-22
>
> Thank you for taking the time to address my concerns. I'll raise my rating.

---

> > ### Author Response · Authors · 2024-08-23
> >
> > Thank you very much! We are excited to hear that our discussion has led to your continued positive evaluation of the work. We will update the paper based on the rebuttal.

---

### Official Review · Reviewer_fPtv · 2024-08-01
**Meaningful contribution to contemporary work**

**Rating:** 7
**Confidence:** 4
**Correctness:** The claims seem sound.
**Clarity:** Yes

**Review:**

Clarity: The paper is fairly easy to follow, there are likely to be several details of experimental setup and data filtering which are missing from the paper but I expect that having access to the papers code will make these aspects clearer.
Quality: The paper is somewhat unpolished and some aspects of the dataset don't follow best practices (e.g. there is no report of manual validation of the data) - however, the construction method seems reasonable enough, and the value of this work to ongoing work is high. Therefore, I lean toward acceptance.
Significance: Clustering tasks is of relevance to the NLP/LLM community and this paper is likely to be useful to ongoing and soon-to-come work.
Originality: The contributions in the paper are novel to the best of my knowledge.

**Strengths:**

- The problem tackled is of relevance to the NLP/LLM community.
- The contributed resource seems fairly well constructed.
- The experiments are reasonable and extensive (though at times lacking in detail).

**Additional Feedback:**

- Did you find any examples of under specified tasks or instructions which could reasonably be interpreted as different tasks? How did you deal with such instructions?
- Please clarify in Table 3 what "Wiki" means? What does "semantic-prompt" mean?

**Documentation:**

A fair amount of detail is provided. The submission also includes code and data.

**Opportunities For Improvement:**

- Please consider adding an experiment with much more detailed appendices detailing the experimental setup for every experiment in 4.3. It should be possible to reproduce your results from reading these appendices.
- Please consider adding a manual validation of the IEB benchmark, even if the manual validation shows the benchmark to be less than perfect knowing the limit of quality in the dataset will help subsequent work.
- It is highly unclear from the paper precisely what a "task" means, this is a challenging question to answer - but I would encourage your work to provide a definition for what you mean by a "task". Conducting the manual analysis may also help arrive at a reasonable definition.

**Relation To Prior Work:**

It is discussed reasonably - some additional suggestions to cite and compare to your work: https://aclanthology.org/2023.findings-acl.225/, https://aclanthology.org/2023.emnlp-main.680/, https://arxiv.org/abs/2005.00770 - you can also consider experimenting with the models of ttps://aclanthology.org/2023.findings-acl.225/ on your dataset.

**Summary And Contributions:**

The paper introduces a dataset of instructions labeled with the task they carry out. The dataset is built by collating instructions in instruction tuning datasets, applying high precision syntactic rules, and then conducting post processing on the resulting instruction clusters using GPT-4 validators and additional cluster merging based on wordnet synonym sets (eg merging "provide" and "give"). Next, the paper explores prompt based instruction embeddings and fine-tuned embedding models to induce instruction clusters. Finally, the paper conducts a series of empirical case studies demonstrating the value of instruction embeddings in various downstream applications.

---

> ### Author Rebuttal · Authors · 2024-08-21
>
> >Examples
>
> We apologize for not fully understanding your question. Are you asking whether there are instructions from specific task categories that could actually represent other tasks? If so, we have found almost no such cases in the existing dataset, as category assignment is only determined by the phrase selected through syntactic analysis. However, theoretically, there could be multi-step instructions where several serialized tasks are included within the same instruction. In such cases, the same instruction might be categorized under different task categories. Since no such cases have been manually identified in the chosen dataset, we haven't separately handled or supplemented these samples. Thanks for your reminding! This could be one of the limitations of our work, which we plan to address in future research.
>
> >Clarification about "wiki" and "semantic-prompt"
>
> We apologize for any ambiguity in our paper and would like to clarify the meanings of "Wiki" and "semantic-prompt" These clarifications will be included in the revised version of the paper. In Table 3, "Wiki" refers to the training set we used for our unsupervised fine-tuned baselines, consistent with the training set employed in SimCSE[1] and PromptBERT[2]. This dataset is sampled from English Wikipedia, hence we use "Wiki" to denote it. Semantic-prompt is a concept introduced in contrast to PIE-prompt. In our paper, the function of PIE-prompt is to guide the model to focus on the tasks within the instructions during encoding, with the specific content of PIE-prompt shown in Figure 6. The term "semantic-prompt" is introduced in contrast to "PIE-prompt". In our paper, the PIE-prompt is designed to guide the model to focus on the tasks specified within the instructions during the encoding process, with the specific content of the PIE-prompt detailed in Figure 6. Conversely, the semantic-prompt aims to direct the model to capture the semantic information of the input text, functioning similarly to the prompt used in PromptBERT[2]. The specific content of the semantic-prompt is presented in Figure 7.
>
> [1] Gao, Tianyu, Xingcheng Yao, and Danqi Chen. SimCSE: Simple Contrastive Learning of Sentence Embeddings. EMNLP 2021.
>
> [2] Jiang, Ting, Jian Jiao, Shaohan Huang, Zihan Zhang, Deqing Wang, Fuzhen Zhuang, Furu Wei, Haizhen Huang, Denvy Deng, and Qi Zhang. Promptbert: Improving bert sentence embeddings with prompts. EMNLP 2022.

---

> ### Author Rebuttal · Authors · 2024-08-21
>
> >Additional related work
>
> Thanks a lot for reminding. The related works you provided are indeed relevant to our work and we will cite them in updated version. We have carefully reviewed them and provided the following analysis:
> 1. TASKWEB: Selecting Better Source Tasks for Multi-task NLP
>
> This paper explores the relationships between NLP tasks and proposes a method for selecting related source tasks based on the target task for model initialization. This approach allows the model, after training on the target task, to achieve better performance than directly fine-tuning on the target task. In our paper, we utilize Instruction Embedding (IE) to encode key task information within instructions. We conduct instruction data selection, benchmark compression based on task diversity, demonstration retrieval based on similar tasks, and an analysis of task correlation ship between instruction sets, validating that our method is applicable to the analysis of instruction-related tasks. Although we did not employ IE to analyze the relationships between instruction tasks, we acknowledge that this is indeed an interesting application of IE. We believe that IE can be used to cluster unannotated instructions, which could then be analyzed for inter-cluster relationships. We plan to investigate this direction further in our future work.
>
> 2. Task-aware Retrieval with Instructions
>
> |Model|ARI|CP|Homo|Silh|IIS-Sp|
> |---|---|---|---|---|---|
> |tart-full-flan-t5-xl|0.2850|0.4469|0.6593|0.1035|0.4018|
> |tart-dual-contriever-msmarco|0.4984|0.6633|0.7994|0.1061|0.7592|
> |Wiki w/o prompt BERT|0.4741|0.6187|0.7741|0.1225|0.7460|
> |EFT-train PIE-prompt BERT **(ours)**|0.8974|**0.9453**|**0.9721**|**0.5180**|0.8446|
> |EFT-train PIE-prompt Llama2 **(ours)**|**0.9125**|0.9432|0.9697|0.4803|**0.8450**|
>
> We experimented with the models from "Task-aware Retrieval with Instructions" on our dataset, and the results are presented above. Since tart-dual-contriever-msmarco is also BERT-based, we compared it with our BERT-based models for detailed analysis. According to the results, tart-dual-contriever-msmarco still falls within the category of semantic embedding, as its performance is similar to that of unsupervised fine-tuned BERT. We attribute this to the domain gap between TART and IE: TART is designed to retrieve target documents based on the instruction task and query content. As a result, instruction task information alone is insufficient for this purpose, necessitating the encoding of semantic information from the query into the TART embedding. In other words, while TART is task-aware, it still incorporates essential semantic information, which can divert its focus from the instruction task when evaluated with our benchmark. In contrast, IE is more focused on the instruction task and thus performs better on our benchmark. However, since IE relies solely on the instruction as input and disregards semantic information, it cannot be directly applied to Information Retrieval tasks.
>
> 3. Exploring and Predicting Transferability across NLP Tasks
>
> The concept of task embedding is closely related to our instruction embedding. However, there is a significant difference between them: In task embedding, the task associated with the data is known in advance, and the embedding is created based on the entire dataset, representing the specific knowledge required for that task. In contrast, with instruction embedding, the task associated with the instructions is unknown beforehand, and the embedding is generated based on a single instruction to represent its intention.

---

> ### Author Rebuttal · Authors · 2024-08-21
>
> Thank you for the valuable and detailed review! We understand your concerns, which are also very important to us. Below we clarify:
>
> >Details about experimental setup
>
> Due to space constraints, we have included the configurations of embedding fine-tuning and Section 4.3.1 in the appendix. Additionally, as you said, more details can be found by checking the code we have released. Below, we provide a more detailed experimental settings related to Section 4.3: For 4.3.1, we complete instruction fine-tuning on 8 NVIDIA A100 (40G) GPU to fine-tune the LLM with full training. The batch size is set to 8 (per device is 1) and gradient accumulation steps are set to 16. The learning rate set to 2 * 10−5 with warmup ratio 0.03. The Alpaca-style template is applied to concatenate queries and responses during fine-tuning. For the inference hyper-parameters in 4.3.2, the decoding temperature is 0.1, the threshold for top p truncation is 0.75 and for top k is 4, and the number of beams is 4. For 4.3.3, similarly to 4.3.1, we use k-means clustering through embeddings to divide the AlpacaEval test set into 10, 50, 100 clusters respectively. For 4.3.4, all the details have been shown in the paper.
>
> >Manual validation
>
> We agree that manual validation is important. In fact, we have already conducted a human evaluation, which is included in the Appendix about Benchmark. We show it here as follows: While we have highlighted the quantity and diversity of the data in IEB, the quality remains uncertain. To assess this, we randomly select 100 task categories and choose one instance from each. Following [1], an expert annotator, who is a co-author of this work, then evaluate whether each instruction belongs to its annotated category. The results indicate that 93\% of the sample categories are accurate. Although we employed various strategies to improve quality, without a comprehensive human check, we can't guarantee that every case is perfectly suitable. However, the results are already quite acceptable.
>
> [1] Yizhong Wang, Yeganeh Kordi, Swaroop Mishra, Alisa Liu, Noah A. Smith, Daniel Khashabi,
> and Hannaneh Hajishirzi. Self-instruct: Aligning language models with self-generated
> instructions. ACL 2023.
>
> >Definition for "task"
>
> Thanks for your advice! We define task as follows: A task of an instruction is a category of activities or work that we expect the LLM to perform, which can be represented by a key phrase (mostly verb-noun phrases). The definition of task is not influenced by specific content or knowledge. For example, "writing an article" is a task, but the specific topic of the article is not important.

---

> ### Author Response · Authors · 2024-08-27
> **Kindly Reminder**
>
> Thank you again for your constructive suggestions! We hope our response adequately addresses your concerns. Please let us know if you have any further questions, as we are happy to continue the discussion.

---

### Decision · Program_Chairs · 2024-09-26

**Decision:**

Accept (Poster)

**Comment:**

This paper defines a novel task to evaluate how well embedding models can identify the underlying task given instructions.
This is a novel and interesting task with a good analysis and an interesting way to create a benchmark. While there were some concerns about the evaluation, the idea is certainly quite novel and interesting.